# PD-L1- and IL-4-expressing basophils promote pathogenic accumulation of T follicular helper cells in lupus

John TCHEN[1,2,9], Quentin SIMON [1,2,3,9], Léa CHAPART [1,2],
Morgane K. THAMINY [1,2], Shamila VIBHUSHAN [1,2], Loredana SAVEANU [1,2],
Yasmine LAMRI[1,2], Fanny SAIDOUNE[1,2], Emeline PACREAU[1,2],
Christophe PELLEFIGUES [1,2], Julie BEX-COUDRAT[1,2], Hajime KARASUYAMA [4],
Kensuke MIYAKE [4], Juan HIDALGO [5], Padraic G. FALLON [6],
Thomas PAPO[1,2,7], Ulrich BLANK [1,2], Marc BENHAMOU[1,2],
Guillaume HANOUNA[1,2,8], Karim SACRE [1,2,7], Eric DAUGAS[1,2,8] &
Nicolas CHARLES [1,2] ✉

Systemic lupus erythematosus (SLE) is an autoimmune disease characterized by anti-nuclear autoantibodies whose production is promoted by autoreactive T follicular helper (TFH) cells. During SLE pathogenesis, basophils accumulate in secondary lymphoid organs (SLO), amplify autoantibody production and disease progression through mechanisms that remain to be defined. Here, we provide evidence for a direct functional relationship between TFH cells and basophils during lupus pathogenesis, both in humans and mice. PD-L1 upregulation on basophils and IL-4 production are associated with TFH and TFH2 cell expansions and with disease activity. Pathogenic TFH cell accumulation, maintenance, and function in SLO were dependent on PD-L1 and IL-4 in basophils, which induced a transcriptional program allowing TFH2 cell differentiation and function. Our study establishes a direct mechanistic link between basophils and TFH cells in SLE that promotes autoantibody production and lupus nephritis.

Systemic lupus erythematosus (SLE) is a multifactorial autoimmune disease that can affect different organs including joints, skin, or kidneys (lupus nephritis)[1]. A break in tolerance leads to an accumulation of autoreactive B and T cells in SLE patients that drives the production of autoreactive antibodies mainly raised against nuclear antigens, such as double-stranded DNA (dsDNA) or ribonucleoproteins (RNP)[2]. These autoantibodies form circulating immune complexes (CIC) with complement factors and autoantigens. CIC deposit in target organs where they can induce a chronic inflammation leading to tissue damage and organ dysfunction[2]. In parallel, CIC activate innate immune cells such as plasmacytoid dendritic cells (pDC), monocytes/macrophages, neutrophils, and basophils, which enhance autoantibody production through the release of pro-inflammatory mediators and initiate a deleterious amplification loop of the disease[1,2].

Autoantibody production is a key event in lupus pathogenesis. Its disruption represents an intensive area of clinical development of

[1]Université Paris Cité, Centre de Recherche sur l'Inflammation, INSERM UMR1149, CNRS EMR8252, Faculté de Médecine site Bichat, 75018 Paris, France. [2]Université Paris Cité, Laboratoire d'Excellence Inflamex, 75018 Paris, France. [3]Inovarion, 75005 Paris, France. [4]Inflammation, Infection and Immunity Laboratory, TMDU Advanced Research Institute, Tokyo Medical and Dental University (TMDU), Tokyo, Japan. [5]Universidad Autonoma de Barcelona, Facultad de Biociencias, Unidad de Fisiologia Animal Bellaterra, Bellaterra Campus, 08193 Barcelona, Spain. [6]School of Medicine, Trinity College Dublin, Dublin 2, Ireland. [7]Service de Médecine Interne, Hôpital Bichat, Assistance Publique – Hôpitaux de Paris, 75018 Paris, France. [8]Service de Néphrologie, Hôpital Bichat, Assistance Publique – Hôpitaux de Paris, 75018 Paris, France. [9]These authors contributed equally: John TCHEN, Quentin SIMON. ✉e-mail: nicolas.charles@inserm.fr

biotherapies targeting directly the B cells or pathways promoting their survival and maturation[3]. Recent advances in the understanding of SLE pathogenesis have shed some light on the role of T follicular helper CD4[+] (TFH) cells and type-2 immunity in addition to tissue damages driven by type 1 and type 17 immunities[2]. TFH cells are central for follicular B cell maturation into antibody-secreting cells and their numbers and functions are dysregulated in both human SLE patients and some lupus-like murine models[4–8]. TFH cells are characterized by the expression of the transcription factor (TF) BCL6 and surface expression of C-X-C motif chemokine receptor 5 (CXCR5), which allows their localization into the germinal centers to provide B cell help through IL-21 production and promote the maturation of antibody-secreting cells[7]. TFH cells express programmed cell-death 1 receptor (PD-1), which is essential for their positioning, functions, and regulation in secondary lymphoid organs (SLO)[9]. GATA3 TF expression, IL-4 production, and potent help provided to B cells characterize the TFH type 2 (TFH2) cell subset, which is overrepresented in the context of lupus and associated with disease activity[6]. TFH cell differentiation are finely tuned through the expression of TF and repressors that regulate their function. Among these, Bcl6 is seen as the lineage-defining TF for TFH cells and represses the *Prdm1* gene (encoding the TF, Blimp1), which is a key TF for effector CD4[+] T cell differentiation[10]. However, a recent study suggested that Blimp1 expression by TFH cells favors their ability to induce plasma cell differentiation and maintenance[11]. Maf and Batf are key TF that regulates the early TFH cell differentiation and their ability to produce IL-21 and IL-4[10]. Gata3 is known to control IL-4 production in TH2 cells and is described as being expressed in TFH2 cells associated with Ets1-deficiency-dependent SLE-like autoimmunity in mice[10,12]. Bach2 is a key TF that represses Blimp1, IL-4, IL-21, Cxcr5, and Bcl6 expressions and competes with TF complexes containing Batf[10]. As a consequence, Bach2 deficiency leads to spontaneous expansion of IL-4 producing TFH2 cells and autoimmunity[13].

Basophils are activated during SLE pathogenesis and accumulate in SLO where they promote autoantibody production by supporting antibody-secreting plasmablast accumulation[14,15]. Depleting basophils in lupus-like mouse models with established disease dampens lupus-like activity by reducing plasmablast numbers, autoantibody titers, CIC deposits in glomeruli, and kidney inflammation[14,15], thus demonstrating that basophils are responsible for an amplification loop driving the disease to a pathogenic threshold. In agreement, constitutive basophil deficiency prevents lupus-like disease onset in the pristane-induced lupus-like disease model[16]. Autoreactive IgE titers in SLE patients are associated with disease activity, increased basophil activation (CD203c), basophil migration abilities (increased CD62L, PTGDR-2, and CXCR4 expressions), and basopenia that reflects their accumulation into SLO[14,15,17–22]. The latter is driven through a prostaglandin D2 (PGD2)-induced CXCR4 externalization on basophil surface, making them sensitive to SLO-derived CXCL12 gradient[15]. In this context, targeting the addressing of basophils to SLO through dual antagonism of PGD2 receptors has demonstrated promising therapeutic potential[15,21] as did the anti-IgE approach[23]. In the *Lyn*[−/−] lupus-like mouse model, we previously showed that basophils accumulate in SLO, where they express surface molecules that suggest interactions with the B and T cell compartments to promote plasma cell maturation and antibody secretion[14]. However, the mechanisms by which basophils, once in SLO, promote this disease amplification loop remains unknown.

In this study, we screened cell surface molecules on blood basophils from SLE patients that could support a functional interaction of these cells with SLO cell partners in lupus. We demonstrate ex vivo in humans and in vivo in two different SLE-like mouse models that PD-L1 expression and IL-4 production by basophils are required to promote their cross-talk with TFH cells to sustain TFH cell pathogenic accumulation, TFH2 cell differentiation, and thereby mediate SLE disease onset and progression.

## Results

### Human blood basophils from SLE patients overexpress PD-L1

TFH, and especially TFH2, cells are significant contributors to the physiopathology of SLE. Proportions among CD4[+] T cells of circulating TFH (cTFH) and cTFH2 cells are increased in several SLE patient cohorts and lupus-like mouse models[4–7,12,24]. We first assessed these cTFH cells (CD3[+] CD4[+] CXCR5[+] ICOS[+] PD-1[+] cells) and cTFH2 cells (CCR6[−] CXCR3[−] TFH cells) in our SLE patient cohort (Supplementary Table 1) and confirmed both their increased representation among CD4[+] T cells, an increased number of cTFH2 cells in active patients despite the SLE-related CD4[+] T cell peripheral lymphopenia[25], and a correlation between cTFH2 cell numbers and disease activity (Spearman $r = 0.3682$, $P = 0.02$, $N = 37$) unlike cTFH1 cell (CCR6[−] CXCR3[+] TFH cells) and cTFH17 cell (CCR6[+] CXCR3[−] TFH cells) proportions[25,26] (Fig. 1a–f and Supplementary Fig. 1a–c).

We and others reported that peripheral basopenia and activation of blood basophils correlate with disease activity in SLE patients[14,15,20,22,27]. Our current SLE patient cohort further confirmed this blood basophil phenotype as evidenced by basopenia and overexpression of the activation marker CD203c, the chemokine receptor CXCR4, and the L-selectin CD62L, all of them being associated with disease activity (Supplementary Fig. 1d–j and Fig. 1g).

To investigate whether, in the context of SLE, human basophils were more prone to interact with TFH cells, we assessed the expression levels of several surface markers identified to be relevant interacting molecules with TFH cells. PD-L2, ICOSL, and OX40L were barely detected on the surface of human blood basophils and no difference in their expression levels on basophils from healthy donors or SLE patients was observed (Supplementary Table 2). Nevertheless, PD-L1 was strongly upregulated on the surface of SLE patient blood basophils as compared to healthy donor blood basophils, independently of the activity of the disease (Fig. 1h and Supplementary Fig. Table 2). In addition, PD-L1 expression on SLE patient blood basophils was positively associated with the basophil activation status (CD203c expression level) (Spearman $r = 0.3113$, $P < 0.0001$, $n = 204$) (Supplementary Fig. 1j). CD84, a member of the SLAM family of proteins important for TFH cell regulation[28], was highly expressed by human basophils (Supplementary Fig. 1k and Supplementary Table 2). Although no significant increase in CD84 expression on blood basophils from the whole SLE patient cohort was found, its expression was significantly increased on blood basophils from active SLE patients as compared to healthy volunteers (Supplementary Fig. 1k and Supplementary Table 2).

Altogether, these results demonstrated that basophils from SLE patients overexpress PD-L1 and CD84, suggesting that a basophil-TFH cell interaction may occur during lupus pathogenesis.

### Basophil-TFH cell axis in lupus-like mouse models

We next sought to verify whether this increased expression of PD-L1 on basophils from SLE patients was also detected on basophils from lupus-like mouse models in which basophil contribution to disease activity is established[14,15,29]. PD-L1 expression was increased on the surface of basophils from both pristane-induced and *Lyn*[−/−] lupus-like mouse models in the analyzed compartments (blood, spleen, and lymph nodes) (Fig. 2a, b). As shown in other human studies and previous reports[7,15], increased TFH cell proportions among CD4[+] T cells and basophil accumulation were observed in both lupus-like mouse models in blood, spleen, and lymph nodes (Fig. 2c–f and Supplementary Fig. 2). These results suggest an association between basophil and TFH cell accumulations in SLO during lupus-like disease.

To evaluate whether a functional relationship between basophils and TFH cells was taking place during the course of the disease, we depleted basophils selectively through diphtheria toxin injection in *Mcpt8*[DTR] mice in both lupus-like models during ten days before their analysis (Supplementary Fig. 2a–c, j, l). We previously showed that basophil depletion dampens CD19[+]CD138[+] short-lived plasma cell

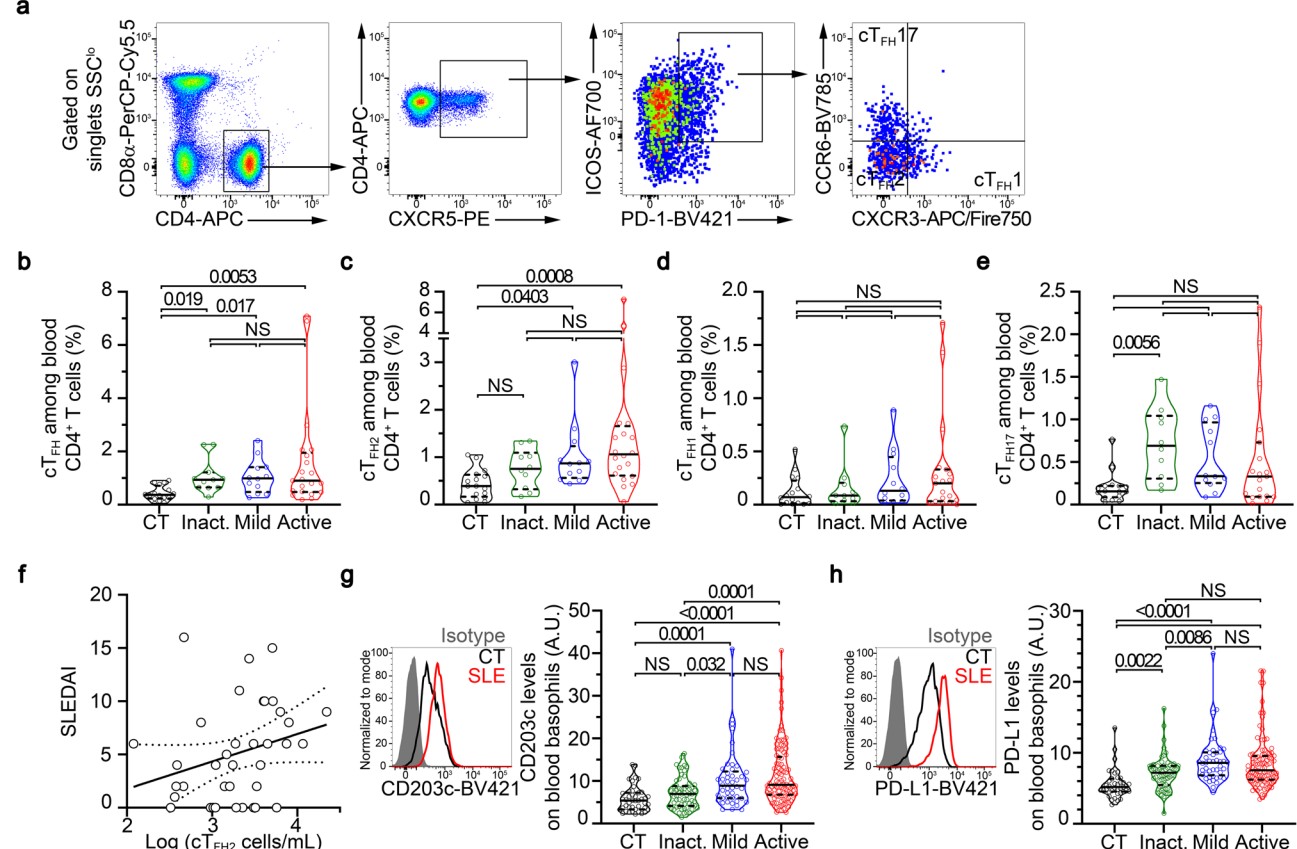

**Fig. 1 | Human blood basophils from SLE patients overexpress PD-L1. a** Flow cytometry gating strategy used to identify human circulating T follicular helper T cells (cTFH) defined as CD3⁺ CD8α⁻ CD4⁺ CXCR5⁺ ICOS⁺ PD-1⁺. Among cTFH cells, cTFH1 cells were defined as CXCR3⁺CCR6⁻, cTFH2 as CXCR3⁻ CCR6⁻, and cTFH17 as CXCR3⁻ CCR6⁺. **b** Proportions (%) among CD4⁺ T cells of cTFH as defined in (**a**) in blood from healthy controls (CT) and inactive (inact.), mild, or active SLE patients (*n* = 16/11/12/21) as determined by flow cytometry. Proportions (%) among CD4⁺ T cells of cTFH2 (**c**), cTFH1 (**d**) and cTFH17 (**e**) as defined in (**a**) in blood from healthy controls and inactive, mild, or active SLE patients (*n* = 16/10/12/20) as determined by flow cytometry. **f** Correlation (and linear regression with 95% confidence intervals) between Log (cTFH2 cell numbers per mL of blood) and SLEDAI (Spearman *r* = 0.3682, *p* = 0.02, n = 37). **g** Left, Histogram plot representing the CD203c

expression levels on blood basophils from a healthy control (CT, black), a patient with active SLE (red), and isotype control staining (gray filled). Right, CD203c expression levels on blood basophils from CT and inact, mild or active SLE patients (*n* = 43/61/46/98) as determined by flow cytometry. **h** Left, Representative histogram plot of PD-L1 expression levels on blood basophils as in (**g**). Right, PD-L1 expression levels on blood basophils from CT and inact, mild or active SLE patients (*n* = 39/60/45/96) as determined by flow cytometry. (**b–e, g, h**) Data are presented as violin plots with median (solid line) and quartiles (dotted lines). Statistical analyses were Kruskal–Wallis tests followed by Dunn's multiple comparisons tests and *p* values are shown above each bracket. NS not significant. A.U. arbitrary units. Source data are provided in the Source Data file.

numbers in SLO, autoantibody titers, kidney deposits and inflammation, and disease activity in these models[14,15,29]. Here, in agreement, efficient DT-mediated basophil depletion (Fig. 2e, f and Supplementary Fig. 2a–c, j, l) resulted in a dramatic decrease in autoreactive IgG antibody titers in both pristane-induced and *Lyn⁻/⁻* lupus-like mouse models (Fig. 2g, h). In addition, it abrogated the increase in TFH cell proportions in blood and SLO from lupus-like mice (Fig. 2c, d, and Supplementary Fig. 2d–f, i, k), strongly suggesting a functional relationship between the two cell compartments. Of note, basophil depletion in *Lyn⁻/⁻ Mcpt8^DTR* mice did not significantly alter the proportion of regulatory TFH (TFR) cells among the CD4⁺ T cell population (Supplementary Fig. 2g, h).

We recently showed that basophils play a nonredundant role in pristane-induced lupus-like disease and that basophil-deficient mice (*Mcpt8^CT/+ Rosa26^DTA/+* mice) are resistant to lupus-like disease onset 8 weeks after pristane injection[16]. The functional relationship between basophils and TFH cells was further confirmed in this mouse model since no increase in the TFH cell proportion was observed in pristane-injected basophil-deficient mice as compared to control mice (*Mcpt8^CT/CT Rosa26^+/+*) in any of the analyzed compartments (Fig. 2i). Moreover, basophil-deficiency still protected the mice from lupus-like

disease 24 weeks after pristane injection, including the TFH accumulation in SLO (Supplementary Fig. 3).

The functional interaction between the two cell types seemed restricted to the lupus-like environment. Indeed, basophil depletion in wild-type (WT) animals in both models or constitutive basophil deficiency did not modify the basal proportions of TFH cells in the observed compartments (Fig. 2c, d, i and Supplementary Fig. 3d, g). Moreover, basophils, which are not involved in the humoral response to the ovalbumin (OVA) protein after intraperitoneal immunization in alum[30], did not influence the rise in TFH cell proportions in OVA-immunized WT *Mcpt8^DTR* mice (Supplementary Fig. 4a–c). The induction of TFH cells in OVA-immunized mice was not associated with the accumulation of basophils in the spleen nor in the draining (mesenteric) lymph node, as previously described by ref. 30, but was associated with the production of anti-OVA IgG antibodies (Supplementary Fig. 4d–g and Supplementary Fig. 5a–c). DT-mediated basophil depletion starting the last 48 h of the immunization procedure did not modify the TFH cell response nor the anti-OVA IgG titers in OVA-immunized WT *Mcpt8^DTR* mice, confirming that basophils were not involved in the support of TFH cells in a classical protein-immunization setting. On the contrary, in a lupus-like model (*Lyn⁻/⁻ Mcpt8^DTR* mice), where basophils are accumulated in SLO

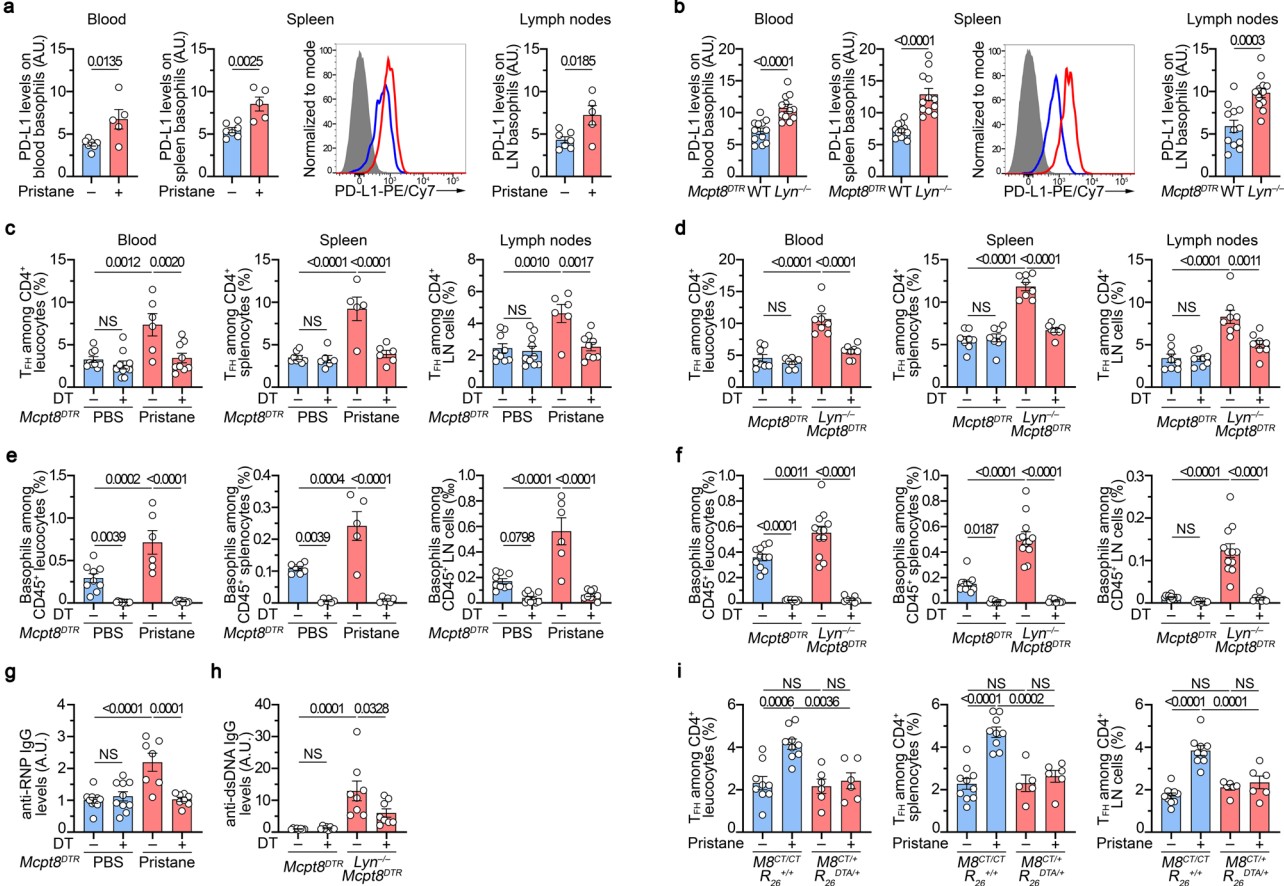

**Fig. 2 | Basophil-TFH functional relationship during lupus-like disease. a** PD-L1 expression levels on basophils from PBS- (blue, $n = 7$) or pristane-injected $Mcpt8^{DTR}$ mice (red, $n = 5$) in the indicated compartments (LN: lymph nodes) (as described in Supplementary Fig. 2a–c). Histograms of PD-L1 expression on spleen basophils from one mouse per group and isotype control (gray filled) are shown. **b** PD-L1 expression levels on basophils from aged $Mcpt8^{DTR}$ (blue, $n = 12$) or $Lyn^{−/−}Mcpt8^{DTR}$ mice (red, $n = 12$) as in (**a**). **c** Proportions (%) of TFH among CD4$^+$ T cells in the indicated compartments from PBS-injected (blue) and basophil sufficient (DT−) or basophil-depleted (DT +) mice and from pristane-injected $Mcpt8^{DTR}$ mice (red) DT treated or not (gating strategy described in Supplementary Fig. 2d–f). Blood ($n = 9/10/6/9$), Spleen ($n = 8/6/5/6$), LN ($n = 9/10/6/9$). **d** Proportions of TFH among CD4$^+$ T cells from aged $Mcpt8^{DTR}$ (blue) and basophil sufficient (DT−) or basophil-depleted (DT +) mice and from aged $Lyn^{−/−}Mcpt8^{DTR}$ mice (red) DT-treated or not, determined as in (**c**) ($n = 8/8/8/8$). **e, f** Proportions of basophils among CD45$^+$ cells in the indicated compartments in mice described in (**c**) and (**d**), respectively.

**e** Blood ($n = 9/10/6/8$), Spleen ($n = 7/6/5/6$), LN ($n = 9/10/6/8$). **f** Blood ($n = 11/8/12/8$), Spleen and LN ($n = 12/8/12/8$). **g** Anti-RNP IgG plasma titers in mice as in (**c**) were quantified by ELISA. O.D. values at 450 nm were normalized to the mean of PBS-injected DT − $Mcpt8^{DTR}$ values ($n = 10/10/7/8$). **h** Anti-dsDNA IgG plasma titers in mice as in (**d**) were quantified by ELISA and normalized to the mean of DT − $Mcpt8^{DTR}$ values ($n = 8/8/8/8$). **i** Proportions of TFH among CD4$^+$ T cells in the indicated compartments from basophil-sufficient ($Mcpt8^{CT/CT}R26^{+/+}$) (blue) or basophil-deficient ($Mcpt8^{CT/+}R26^{DTA/+}$) (red) mice treated with PBS (−) or pristane (+). Blood ($n = 9/9/6/6$), Spleen ($n = 10/9/5/6$), LN ($n = 10/9/6/6$). **a–i** Results are from at least three independent experiments and presented as individual values in bars representing the mean values ± s.e.m. Statistical analyses were done by two-sided unpaired $t$ test (**a**, **b**) or by two-way ANOVA followed by Tukey's multiple comparisons test (**c–i**) between the indicated groups, and $p$ values are shown above each bracket. NS: not significant. A.U.: arbitrary units. Source data are provided in the Source Data file.

(Fig. 2, Supplementary Figs. 4d, e and 5e, f and refs. [14],[15]), basophil depletion decreased proportions of TFH cells resulting in reduced anti-OVA IgG antibody production (Supplementary Fig. 4g). Thus, this functional relationship seemed to be linked to the accumulation of basophils in SLO observed in the lupus-like context. We previously showed that intraperitoneal PGD$_2$ injection in mice was leading to a CXCR4-dependent recruitment of basophils into SLO[15]. We next induced basophil accumulation in SLO in WT $Mcpt8^{DTR}$, $Mcpt8^{CT/+}$, and $Mcpt8^{CT/+}$ $Rosa26^{DTA/+}$ mice by injecting PGD$_2$ to the mice every three days during the whole OVA immunization procedure (Supplementary Fig. 4a). As expected, this led to the accumulation of basophils in spleen and draining (mesenteric) lymph node. The accumulation of basophils occurred mainly at the T cell:B cell (T:B) border in the spleen, and resulted in increased germinal center (GC) formation as compared to non-PGD$_2$ treated OVA-immunized mice (Supplementary Figs. 4 and 5a–d). Importantly, promotion of GC formation was not observed in PGD$_2$-treated OVA-immunized basophil-deficient animals further

showing the basophil contribution to this phenomenon (Supplementary Fig. 5a–d). Mimicking the lupus-like context through basophil accumulation in SLO, PGD$_2$ treatment made the TFH expansion and the resulting anti-OVA IgG production dependent on basophils as shown after DT-induced basophil depletion (Supplementary Fig. 4). Of note, spleen basophils of pristane-treated animals were as well located mainly at the T:B border and were needed to observe increased GC-like structure accumulation and lupus-like disease onset (Supplementary Fig. 5e–g).

Altogether, these data strongly suggest that basophils and TFH cells share a functional relationship in the lupus-like context. Indeed, once accumulated in SLO, basophils enable the expansion of the TFH cell population that controls the contextual humoral response.

**Basophils control TFH cell ability to produce IL-21 and IL-4 in lupus-like models**

TFH cells promote B cell maturation (isotype switch, affinity maturation, and differentiation towards antibody-secreting cells) inside (CD90.2$^−$

TFH cells) and outside (CD90.2[+] TFH cells) GC by providing IL-21 and IL-4 to their environment and by interacting with activated B cells especially at the T:B border and in GC in SLO[31,32]. We next assessed whether basophils, beyond their effects on TFH cell numbers in the lupus-like context, could influence the functions of the expanded TFH cells during the disease.

First, the proportions of spleen TFH cells producing IL-21 without any restimulation were significantly increased in both lupus-like models and further amplified after phorbol myristate acetate (PMA) and ionomycin restimulation as compared to spleen TFH cells from control mice (Fig. 3a, b and Supplementary Fig. 6a). This suggested that in the lupus-like context, TFH cells are more prone to provide IL-21 to surrounding cells in these mouse models. Second, the same observations were done concerning IL-4 production by TFH cells showing a TFH2 bias in these two lupus-like models (Fig. 3c, d and Supplementary Fig. 6b), as also observed in the SLE patient cohort (Fig. 1c and Supplementary Fig. 1c). No such constitutive bias was observed for IFNγ production by TFH cells in none of the two mouse models and a basophil-independent TFH17 constitutive bias was observed only in the aged Lyn[−/−]xMcpt8[DTR] mice (Fig. 3e, f and Supplementary Fig. 6c, d). Third, basophil depletion dramatically dampened both constitutive and PMA-ionomycin-induced IL-21 and IL-4 productions by TFH cells only in the lupus-like context, without significantly influencing their IFNγ or IL-17A production abilities

(Fig. 3a–f and Supplementary Fig. 6a–d). Of note, the analysis of non-TFH CD4[+] T helper cells (defined as CD45[+] CD19[−] TCRβ[+] CD8α[−] CD4[+] CD44[+] CXCR5[−] cells) revealed that basophil depletion in lupus-like models impacted only the TH2 cell compartment in both mouse models (Supplementary Fig. 6e–k). These results strongly suggest that, in the lupus-like context, basophils promoted TFH cell-derived IL-21 and IL-4 production ability and were responsible for a TFH2 cell bias of this T cell population.

In line with the observed effects on TFH cell numbers and cytokine production, basophil depletion led to decreased switched B cell proportions in Lyn[−/−]xMcpt8[DTR] mice (Fig. 3g, h). Purified TFH cells from these basophil-depleted mice were less efficient than basophil-sufficient mice to induce WT B cell differentiation into plasmablast in co-culture experiments (Fig. 3i, j). Finally, if basophil depletion dampened both CD90.2[+] and CD90.2[−] TFH cells, it induced increased proportions of CD90.2[−] TFH cells among TFH cells suggesting that basophils were first promoting CD90.2[+] TFH cells (Fig. 3k–n). Whereas its high expression may also reflect their localization into GC[31], CXCR5 levels on TFH cells were not modified by basophil depletion in both lupus-like mouse models (Fig. 3o, p).

Altogether, these results strongly suggest that in the lupus-like context basophils in SLO enable TFH cell accumulation, increase TFH cell ability to produce IL-21 and IL-4 and promote TFH cell function including B cell isotype switch and plasmablast differentiation.

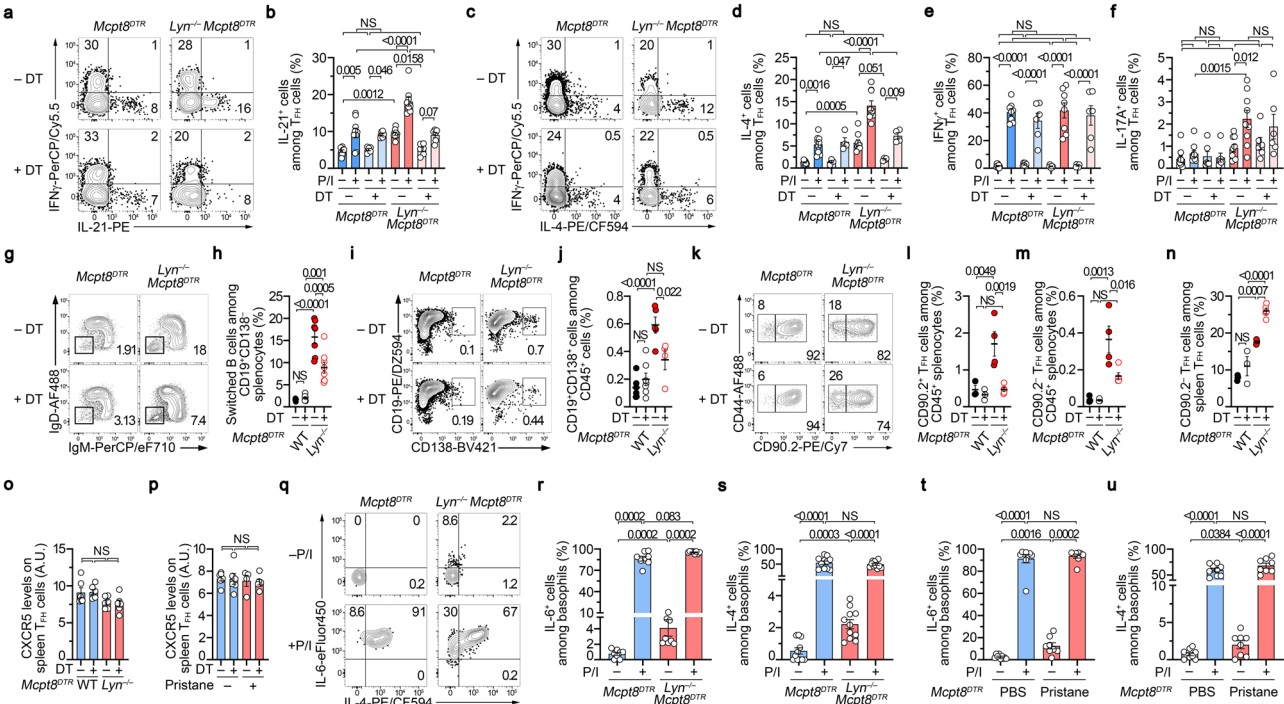

**Fig. 3 | Basophils control TFH abilities to produce IL-21 and IL-4 in the lupus-like context. a, c** Contour plots showing PMA and Ionomycin (P/I)-induced IFNγ, IL-21, and IL-4 productions by TFH cells (as defined in supplementary Fig. 2d-f) in splenocytes from aged Mcpt8[DTR] or Lyn[−/−]Mcpt8[DTR] mice basophil-depleted (DT +) or not (DT−). Proportions (%) of IL-21+ (**b**) (n = 9/6/9/6), IL-4+ (**d**) (n = 9/4/9/4), IFNγ + (**e**) (n = 8/6/8/6) and IL-17A+ (n = 9/6/9/6) (**f**) TFH cells in splenocytes as in (**a**). **g** Contour plots of switched B cells (IgM[−]IgD[−]) among spleen CD45[+]TCRβ[−]CD19[+]CD138[−] cells in mice as in (**a**). **h** Proportions (%) of switched B cells among CD19[+]CD138[−] splenocytes as in (**g**) (n = 6/6/7/8). **i** Contour plots of CD19[+]CD138[+] cells among CD45[+] cells from co-culture of sorted TFH cells from mice as in (**g**) and WT B cells. **j** Proportions (%) of CD19[+]CD138[+] cells among CD45[+] cells as in (**i**) (n = 6/7/5/4). **k** Contour plots of CD90.2[+] and CD90.2[−] TFH cells among spleen TFH cells from mice as in (**a**). Proportions (%) of CD90.2[+] (**l**) and CD90.2[−] (**m**) TFH cells among CD45[+] splenocytes from mice as in (**a**) (n = 3/3/4/5). **n** Proportions (%) of CD90.2[−] TFH cells among spleen TFH cells as in (**k**)

from mice as in (**g**) (n = 3/3/4/5). CXCR5 expression on spleen TFH cells from mice as in (**a–n**) (**o**) and as in (**t, u**) (**p**). **q** Contour plots showing non-stimulated (−) and P/I-induced (+) IL-6 and IL-4 productions by spleen basophils from aged Mcpt8[DTR] or Lyn[−/−]Mcpt8[DTR] mice. Proportions (%) of IL-6+ (**r**) (n = 8/8/8/8) and IL-4+ (**s**) (n = 11/11/11/11) basophils in splenocytes as in (**a**). Proportions (%) of IL-6 (**t**) (n = 9/9/8/8) and IL-4 (**u**) (n = 9/9/8/8) producing cells among basophils in splenocytes stimulated with P/I (+) or not (−) from PBS-injected Mcpt8[DTR] (blue) or pristane-injected Mcpt8[DTR] (red) mice. **b, d–f, h, j, l–p, r–u** Results are from at least three independent experiments and presented as individual values in bars representing the mean values ± s.e.m. Statistical analyses were by two-way ANOVA followed by Tukey's multiple comparisons test (**b, d–f, h, j, l–p**) or by Kruskal–Wallis tests followed by two-sided Mann–Whitney U test between the indicated groups (**r–u**). P values are shown above each bracket. NS not significant. Source data are provided in the Source Data file.

Both IL-6 and IL-4 are involved, respectively, in TFH and TFH2 cell differentiation[7,12]. Murine basophils are potent producers of these cytokines[33,34]. We next assessed whether basophils in both pristane-injected and Lyn-deficient animals were more prone to produce these cytokines with or without PMA-Ionomycin restimulation. Unlike what was observed in basophils from control mice, constitutive IL-4 and IL-6 productions were significantly detected in non-restimulated basophils from both lupus-like models, whereas no major differences were noticed after PMA-ionomycin restimulation (Fig. 3q–u). These results suggested that during the course of the disease, along with their increased expression of PD-L1 (Fig. 2a, b), basophils produce IL-6 and IL-4 in vivo explaining, at least partially, some of their effects on the expansion of the TFH cell population and its TFH2 cell bias.

### Basophils promote ex vivo TFH cell differentiation through IL-4- and PD-L1-dependent mechanisms

Next, we evaluated the effects of basophils and basophil-expressed mediators on the differentiation of TFH cells ex vivo in a co-culture system. The presence of basophils induced a clear differentiation of the CD3/CD28-activated naïve CD4+ T cells towards the TFH cell subset with an increased ability of these Bcl6+ TFH cells to produce IL-21 and IL-6 (Fig. 4a–c and Supplementary Fig. 7a). Moreover, the basophil-induced TFH cell differentiation was biased towards the TFH2 cell subset as evidenced by an increased ability to produce IL-4 and IL-13

and an increased expression of GATA3 (Fig. 4d–f). We next bred CT-M8 (or $Mcpt8^{CT/CT}$) mice[16] with $Il4^{fl/fl[35]}$, $Il6^{fl/fl[36]}$, or $Pdl1^{fl/fl[37]}$ mice and generated mice deficient for IL-4, IL-6, or PD-L1 selectively in the basophil compartment (Supplementary Fig. 7b–d). IL-4-deficient basophils could not induce any of the effects on the TFH cell differentiation as compared to WT basophils (Fig. 4a–e). IL-6-deficient basophils could still induce TFH cell differentiation despite reduced IL-6, but increased IL-21, production by TFH cells (Fig. 4a–c) and had a limited effect on the TFH2 cell differentiation (Fig. 4d, e). Interestingly, PD-L1 expression by basophils was mandatory to induce TFH cell differentiation and had a limited impact on the TFH2 cell differentiation that was mainly dependent on basophil-expressed IL-4 (Fig. 4a–e). Of note, PMA-ionomycin restimulation of the co-cultured cells in the aforementioned conditions led to the same conclusions (Supplementary Fig. 7e–h). The relevance of PD-1/PD-L1 interaction in this process was further validated with the addition of blocking recombinant PD-1-Fc molecules to the co-culture of CD3/CD28-activated naïve CD4+ T cells with WT basophils (CT-M8) that significantly reduced basophil-induced TFH cell differentiation (Supplementary Fig. 7i).

We next sought to identify which TF program was induced by basophils in CD3/CD28 activated naïve CD4+ T cells after 3 days in co-culture. Together with Bcl6 expression, Bach2 downregulation is a key event controlling TFH cell differentiation through Cxcr5 induction

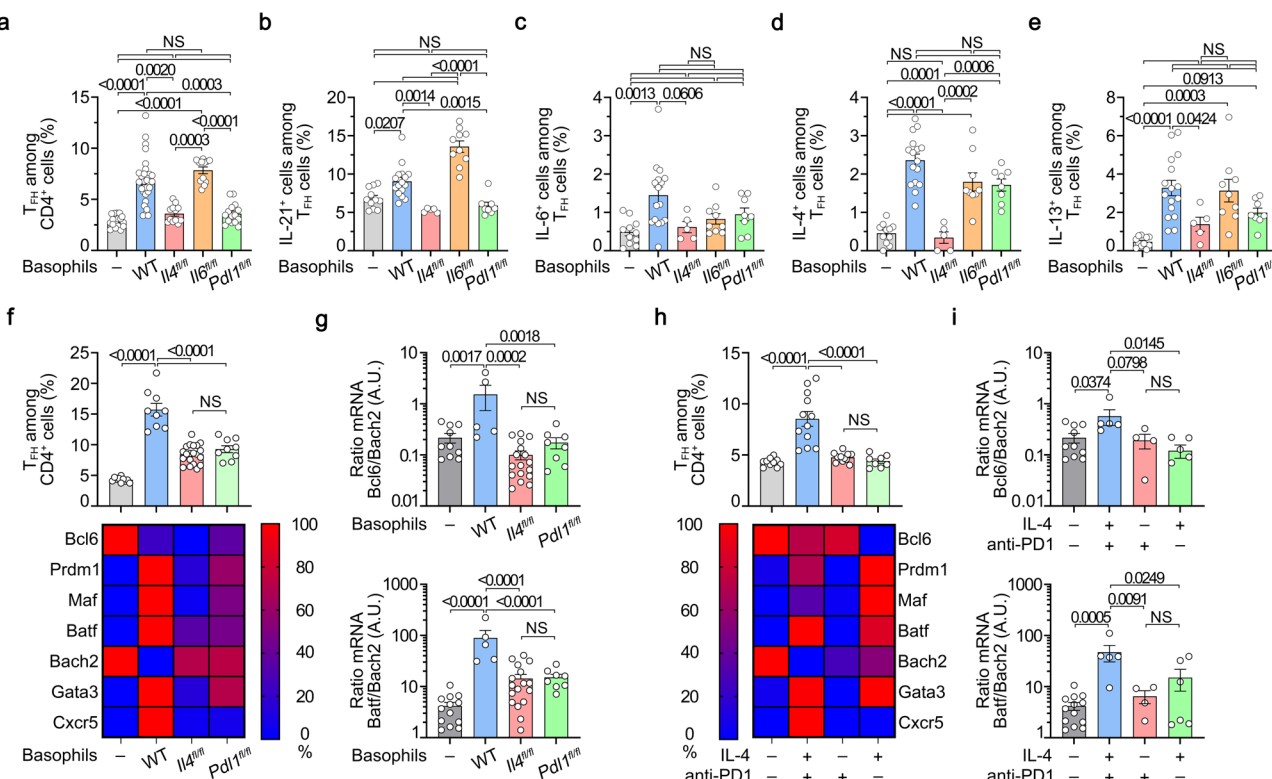

**Fig. 4 | PD-L1- and IL-4-expressing basophils promotes ex vivo CD4+ T cell differentiation into TFH cells. a–g** CD3/CD28-stimulated wild-type (WT) naïve CD4+ T cells cultured for three days without (−, gray) or with basophils from $Mcpt8^{CT/+}$ (WT) (blue), $Mcpt8^{CT/+}Il4^{fl/fl}$ (red), $Mcpt8^{CT/+}Il6^{fl/fl}$ (orange), or $Mcpt8^{CT/+}Pdl1^{fl/fl}$ (green) mice. **a** Proportions (%) of TFH cells among CD4+ T cells ($n = 17/31/12/12/15$). Proportions (%) of IL-21- (**b**), IL-6- (**c**), IL-4- (**d**), and IL-13- (**e**) producing cells among TFH cells non-restimulated ($n = 11/16/5/10/8$). **f** (*Top*) Proportions (%) of TFH cells among CD4+ T cells ($n = 10/9/20/9$). *Bottom* RT-qPCR done for the indicated targets on resorted CD4+ T cells. Relative results are presented with the color scale indicated from 0% (blue, the least abundant) to 100% (red, the most abundant). **g** Ratio of Bcl6 on Bach2 mRNA (*Top*) ($n = 10/5/17/8$) and of Batf on Bach2 mRNA (*Bottom*) ($n = 13/5/17/8$) as described in (**f**). **h** *Top* Proportions (%) of TFH cells among CD4+

T cells cultured without basophils for three days either alone (−, gray) or with coated anti-PD-1 antibody (red) or soluble IL-4 (green) or both (blue) ($n = 10/12/12/8$). *Bottom*. Relative mRNA expression of the indicated targets as in (**f**). **i** Ratio of Bcl6 on Bach2 mRNA (*Top*) ($n = 10/5/4/6$) and of Batf on Bach2 mRNA (*Bottom*) ($n = 13/5/4/6$) on samples described in (**h**). **b–e** Results for PMA/Ionomycin restimulated cells are shown in Supplementary Fig. 7e–h. **a–i** Results are from at least three independent experiments and presented as individual values in bars representing the mean ± s.e.m. Statistical analyses were done by Kruskal–Wallis test followed by Dunn's multiple comparisons tests (**a**) or by one way ANOVA followed by Tukey's multiple comparisons tests (**b–i**) between the indicated groups. P values are shown above each bracket. NS not significant. A.U. arbitrary units. Source data are provided in the Source Data file.

and the promotion of the complex Batf/Irf4 that activates the Bcl6 promoter[38,39]. Consequently, Bach2-deficient mice expand some IL-4 producing TFH cells and develop autoimmunity[13]. Here, in our co-culture system, without basophils, both Bcl6 and Bach2 mRNA were highly expressed by CD4[+] T cells unlike Prdm1, Maf, Batf, Gata3 and Cxcr5. This TF program was associated with very low Bcl6/Bach2 and Batf/Bach2 mRNA ratios resulting in a low proportion of TFH cell differentiation in the culture (Fig. 4f, g). Co-culture with WT basophils dramatically enhanced both Bcl6/Bach2 and Batf/Bach2 mRNA ratios and strongly induced Prdm1, Maf, Gata3, and Cxcr5 mRNA expressions. In these conditions, the repressor Bach2 mRNA was not detected anymore and TFH cell differentiation was strongly induced (Fig. 4f, g). Prdm1, Maf, Batf, and Gata3 mRNA inductions depended mainly on basophil-derived IL-4, whereas Bach2 downregulation, Cxcr5 mRNA, Bcl6/Bach2, and Batf/Bach2 mRNA ratio inductions were dependent on the presence of both basophil-expressed IL-4- and PD-L1-dependent signals (Fig. 4f, g). Similar results to the effects of WT basophils were obtained with both coated anti-PD-1 monoclonal antibody and IL-4 addition to CD3/CD28 activated naïve CD4[+] T cells for three days (Fig. 4h, i). Prdm1, Maf, Batf, and Gata3 mRNA were mainly induced by IL-4 whereas Bcl6 maintenance was dependent on PD-1 engagement on T cells (Fig. 4h, i). Bach2 mRNA downregulation, Cxcr5 mRNA, Bcl6/Bach2, and Batf/Bach2 mRNA ratio inductions were optimally achieved by stimulating CD3/CD28 activated CD4[+] T cells for 3 days with both IL-4 and anti-PD-1 antibody that allowed to induce TFH cell differentiation as WT basophils did (Fig. 4f–i).

Altogether, these results demonstrated that basophils induce ex vivo TFH cell differentiation in an IL-4- and PD-L1-dependent manner by inducing a TF program enabling TFH cell differentiation. The same TF combination was induced by PD-1 engagement on CD3/CD28 activated CD4[+] T cells in the presence of IL-4.

## CD4[+] T cells promote basophil ability to induce TFH cell differentiation ex vivo

In parallel, we analyzed the effects of the CD4[+] T cells on the basophil compartment. The presence of CD3/CD28 activated CD4[+] T cells dramatically up-regulated PD-L1 expression on basophils (Fig. 5a, b), as observed in vivo in the lupus-like context (Fig. 2). This effect was reduced when basophils were IL-4-deficient (Fig. 5a, b). IL-3 is known to potently activate basophils, its production by T cells is known to be upregulated in SLE patients' blood as well as in lupus-like mouse models and SLE patient IL-3 signature is associated with *CD274* (PD-L1) gene upregulation in whole blood[40–46]. Autoreactive IgE are as well described as basophil-activating factors in SLE[47]. Thus, we analyzed whether IL-4, IL-3 and/or FcεRI-crosslinking could induce PD-L1 upregulation on the surface of basophils. We first confirmed that CD3/CD28 activated CD4[+] T cells in the co-culture system were producing IL-3 in a basophil-independent manner (Fig. 5c). Addition of IL-3 to splenocytes ex vivo increased PD-L1 expression on the surface of basophils as did FcεRI crosslinking (Fig. 5d) suggesting that these signals may contribute to PD-L1 overexpression on basophils in the lupus context (Figs. 1 and 2). Of note, IL-4 alone did not induce PD-L1 overexpression but enhanced the one induced by IL-3 whereas IL-3 and FcεRI-crosslinking synergized to further enhance PD-L1 levels (Fig. 5d). The upregulation of PD-L1 on basophils by IL-3 was rapid and dose-dependent (Supplementary Fig. 7j).

CD3/CD28 activated CD4[+] T cells induced constitutive IL-4, IL-6, and IL-13 productions by basophils (Fig. 5e–g and Supplementary

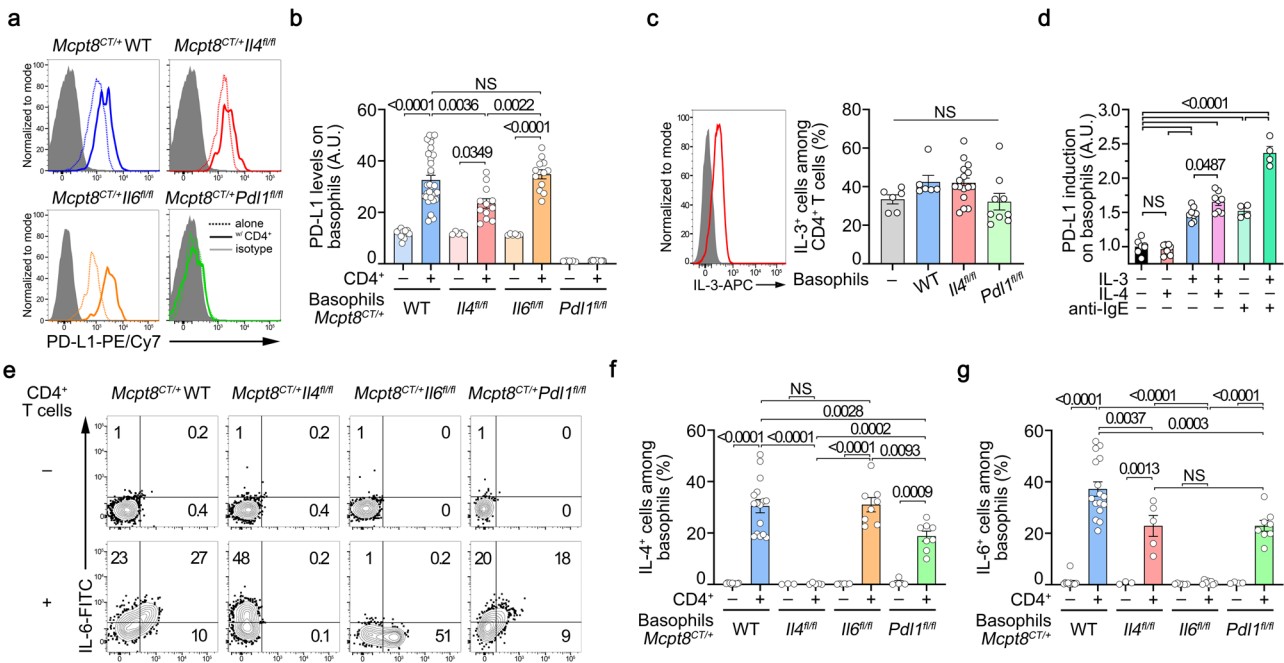

**Fig. 5 | CD4[+] T cells promotes ex vivo PD-L1 and IL-4 expressions by basophils.**
**a** Histograms of PD-L1 expression on basophils cultured without (dotted) or with (solid) activated naïve CD4[+] T cells as in Fig. 4 (gray filled: isotype control staining). **b** PD-L1 expression levels on basophils cultured without (−, lighter colors) or with (+, darker colors) activated naïve CD4[+] T cells (n = 14/31/5/12/6/12/5/12). **c** *Left*. Intracellular IL-3 staining in WT CD3/CD28-activated naïve CD4[+] T cells after 3 days of culture (red line) (gray filled: isotype control staining). *Right*. Proportions (%) of IL-3[+] cells among CD4[+] T cells non-restimulated in the same samples as in Fig. 4f (n = 6/9/16/9). **d** PD-L1 expression induction on WT basophils after stimulation of splenocytes with 1 ng/mL of IL-4 (red), of IL-3 (blue), both (purple), 100 ng/mL of anti-IgE without (light green) or with (green) IL-3 for 20 h (n = 8/8/8/8/4/4)

(normalized to unstimulated conditions mean value). **e** Contour plots of IL-6 and IL-4 spontaneous production by basophils of the indicated genotype after co-culture without (−) or with activated CD4[+] T cells. Proportions (%) of spontaneous IL-4[+] (**f**) and IL-6[+] (**g**) basophils of the indicated genotypes co-cultured (+) or not (−) with WT CD3/CD28-activated naïve CD4 + T cells (n = 11/16/3/5/6/8/4/8). **f, g** Results for IL-13 and PMA/Ionomycin restimulated cells are shown in Supplementary Fig. 7k–n.
**a–g** Results are from at least three independent experiments and presented as individual values in bars representing the mean ± s.e.m. **b–d, f, g** Statistical analyses were by one way ANOVA followed by Tukey's multiple comparisons tests between the indicated groups. P values are shown above each bracket. NS not significant. A.U. arbitrary units. Source data are provided in the Source Data file.

Fig. 7k). The CD4$^+$ T cell-induced up-regulation of PD-L1 on basophils was partially dependent on the IL-4 produced by basophils themselves, but independent of basophil-derived IL-6 (Fig. 5b). Conversely, the CD4$^+$ T cell-induced IL-4 (and IL-6) production by basophils was partially dependent on PD-L1 expressed by basophils without impacting their maximal production ability (Fig. 5e–g and Supplementary Fig. 7l–n). IL-6 deficiency in basophils did not alter their ability to produce IL-4 constitutively in the presence of CD4$^+$ T cells nor after PMA-ionomycin restimulation but enhanced their IL-13 production only in the former situation (Fig. 5e–g and Supplementary Fig. 7k–n). These results suggest that PD-L1 engagement on basophils by CD4$^+$ T cells, beyond promoting IL-21 production by TFH cells (Fig. 4b), was responsible for the extent of T cell-induced cytokine production by basophils. This may explain why PD-L1 deficient basophils could not promote TFH cell differentiation (Fig. 4a). Of note, CD4$^+$ T cells induced IL-13 production by basophils independently of their IL-4 or PD-L1 expression (Supplementary Fig. 7k, n).

Altogether, these results strongly suggest that a bi-directional interaction between naïve CD4$^+$ T cells and basophils promotes the

differentiation of TFH cells via a mechanism mainly depending on IL-4 and PD-L1 expression by basophils but independent of the basophil-derived IL-6.

## PD-L1 controls the basophil-TFH cell functional relationship during lupus-like disease

We next sought to validate in vivo the relevance of PD-L1 expression by basophils in their functional relationship with TFH cells in the lupus-like context. Pristane injection did not lead to TFH cell accumulation in SLO in mice with basophil-restricted PD-L1-deficiency (*Mcpt8*$^{CT/+}$ *Pdl1*$^{fl/fl}$) as compared to WT animals (*Mcpt8*$^{CT/+}$) (Fig. 6a and Supplementary Fig. 8). Moreover, TFH cells from the pristane-injected *Mcpt8*$^{CT/+}$ *Pdl1*$^{fl/fl}$ mice did not produce more IL-21 nor IL-4 than PBS-injected mice as compared to their WT counterparts (Supplementary Fig. 9a, b). However, PD-L1 expression by basophils was not required for basophil accumulation in SLO (Fig. 6b and Supplementary Fig. 8a). Both the lupus-like context in vivo (Fig. 3q–u) and the co-culture system in vitro (Fig. 5e–g) induced IL-4 and IL-6 production by basophils. Here, PD-L1 deficiency on basophils completely prevented these cytokine

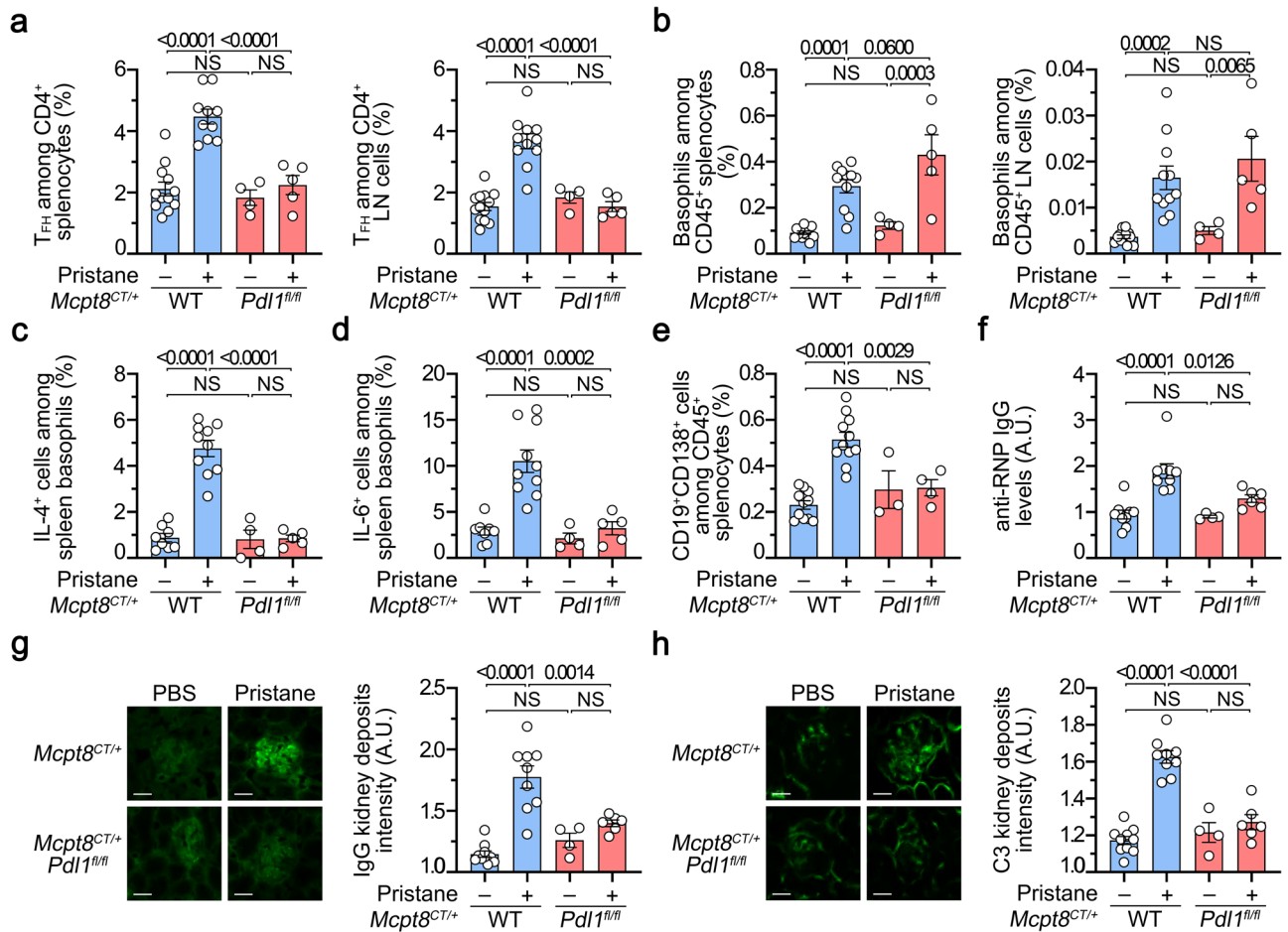

**Fig. 6 | PD-L1 controls the basophil-TFH functional relationship during lupus-like disease. a** Proportions (%) of TFH cells among CD4$^+$ T cells in spleen (left) (*n* = 12/10/4/5) and lymph nodes (LN) (right) (*n* = 13/11/4/5) from *Mcpt8*$^{CT/+}$ (WT) (blue) or *Mcpt8*$^{CT/+}$ *Pdl1*$^{fl/fl}$ (*Pdl1*$^{fl/fl}$) (red) mice injected with PBS (–) or with pristane (+). **b** Proportions (%) of basophils among CD45$^+$ cells in the spleen (left) (*n* = 12/10/4/5) and lymph nodes (right) (*n* = 13/11/4/5) from the mice described in (**a**). Proportions (%) of spontaneous IL-4$^+$ (**c**) (*n* = 8/10/4/5) or IL-6$^+$ cells (**d**) (*n* = 8/10/4/5) among basophils in the spleen from mice as in (**a**). **e** Proportions (%) of CD19$^+$CD138$^+$ cells among CD45$^+$ cells in spleen from the mice described in (**a**) (*n* = 11/11/3/4). **f** Anti-RNP IgG autoantibody plasma titers from the same mice as in (**a**) were quantified by ELISA and data were normalized to the mean of PBS-injected

*Mcpt8*$^{CT/+}$ values (*n* = 9/9/4/6). Left, Representative pictures of one glomerulus from mice with the indicated genotypes treated without (PBS, –) or with pristane (+) showing the intensity of anti-IgG (**g**) or anti-C3 (**h**) staining by immunofluorescence. Scale bar = 50 μm. Uncropped images are shown in Supplementary Fig. 9a (IgG) and 9b (C3). Right, quantification of IgG (**g**) and C3 (**h**) glomerular deposits in kidneys from the mice described in (**a**) (*n* = 10/9/4/6). **a**–**h** Results are from at least three independent experiments and presented as individual values in bars representing the mean values ± s.e.m. Statistical analyses were done by two way ANOVA followed by Tukey's multiple comparisons test between the indicated groups. *P* values are shown above each bracket. NS not significant. Source data are provided in the Source Data file.

productions in vivo in the lupus-like context (Fig. 6c, d). This was associated with a lack of expansion of CD19⁺CD138⁺ cells (Fig. 6e) and of GC B cells (Supplementary Fig. 10) in SLO leading to dramatically reduced anti-RNP autoreactive IgG titers in the blood (Fig. 6f). Moreover, IgG and C3 glomerular deposits were barely detectable in the kidney of pristane-injected $Mcpt8^{CT/+}$ $Pdl1^{fl/fl}$ mice as compared to their WT counterparts (Fig. 6g, h and Supplementary Fig. 9c, d).

Altogether, these results strongly suggest that PD-L1 basophil expression and up-regulation during lupus development were not involved in basophil accumulation in SLO but were responsible for the basophil-induced promotion of TFH cell and short-lived plasma cell expansions and subsequent pathological parameters.

### Basophil-derived IL-4 exerts a dual effect during lupus-like disease development

Since TFH cell differentiation ex vivo depended on IL-4 expression by basophils (Fig. 4a–i), we next sought to verify whether basophil-derived IL-4 was mandatory for the basophil-TFH cell functional relationship in the lupus-like context. As suggested by above results, basophil-specific IL-4 deficiency ($Mcpt8^{CT/+}$ $Il4^{fl/fl}$) prevented the pristane-induced TFH cell accumulation in SLO as compared to WT ($Mcpt8^{CT/+}$) animals (Fig. 7a) and the remaining TFH cells did not produce more IL-21 nor IL-4 as compared to their WT counterparts (Supplementary Fig. 9a, b). However, basophil recruitment into SLO upon pristane treatment was not dependent on basophil-derived IL-4 (Fig. 7b). Surprisingly, CD19⁺CD138⁺ cells still accumulated in SLO despite the selective basophil IL-4 deficiency and the lack of TFH cell and germinal center (GC) B cell accumulations (Fig. 7a, c and Supplementary Figs. 3b and 10). This phenotype was associated with a still significant constitutive IL-6 production by IL-4-deficient basophils in the lupus-like context (Fig. 7d). However, no significant titers of anti-RNP IgG autoantibodies were detected in the blood of pristane-injected $Mcpt8^{CT/+}$ $Il4^{fl/fl}$ mice as compared to their WT counterparts (Fig. 7e). In line with this feature, no IgG deposits were present in the glomeruli of pristane-treated mice with basophil-specific IL-4 deficiency but C3 deposits, although reduced, were still detected (Fig. 7f, Supplementary Fig. 9c, d). Along with the accumulation of CD19⁺CD138⁺ cells in the SLO, the latter result led us to assess the presence of autoreactive IgM in the pristane-treated $Mcpt8^{CT/+}$ $Il4^{fl/fl}$ mice. As suspected, increased titers of anti-RNP IgM in the plasma of mice with IL-4 deficient basophils were observed as compared to pristane-treated $Mcpt8^{CT/+}$ WT mice (Fig. 7g). These autoreactive IgM were not detected in the blood from pristane-injected basophil-specific PD-L1-deficient mice nor constitutive basophil-deficient mice (Fig. 7g) in line with the absence of plasmablast accumulation in SLO from these mice (Fig. 6e and ref. 16). These anti-RNP IgM titers correlated with the presence of IgM deposits in the glomeruli of the corresponding mice (Fig. 7h, i).

Altogether, these results strongly suggest that TFH cell pathogenic accumulation in SLO and autoreactive IgG titers induced by pristane were dependent on the basophil-derived IL-4. However, CD19⁺CD138⁺ short-lived plasma cell accumulation, which is dependent on basophils (Fig. 2 and refs. 15,16,29), was not dependent on IL-4 production by basophils unlike GC B cell accumulation (Supplementary Fig. 10). This led to an accumulation of plasmatic autoreactive anti-RNP IgM, with no IgG deposits but increased C3 deposits in glomeruli of the $Mcpt8^{CT/+}$ $Il4^{fl/fl}$ mice, suggesting that basophil-derived IL-4 was both enabling autoreactive antibody switch towards the IgG isotype and TFH cell and GC B cell accumulations that promoted this phenomenon as well[7,12].

### IL-4 and IL-3 control PD-L1 expression, activation and localization of basophils in the lupus-like context

Since CD19⁺CD138⁺ short-lived plasma cell accumulation depended on PD-L1 expression by basophils in the lupus-like context (Fig. 6), we assessed PD-L1 expression levels on basophils from pristane-treated $Mcpt8^{CT/+}$ $Il4^{fl/fl}$ mice. Although weaker than on WT counterparts, PD-L1 up-regulation on basophils from basophil-specific IL-4 deficient animals was detected (Fig. 8a and Supplementary Fig. 11a), in line with the reduced induction of PD-L1 by CD4 + T cells on IL-4 deficient basophils ex vivo (Fig. 5a, b) and the IL-4-mediated promotion of the IL-3 induced PD-L1 expression ex vivo (Fig. 5d). IL-3 titers is known to be up-regulated in SLE patient serum and in lupus-like mouse models[40–44,46]. Plasmatic IL-3 concentrations were increased in all pristane-treated animals independently of the presence of basophils nor of the basophil-selective IL-4 or PD-L1 deficiency and were also increased in aged $Lyn^{-/-}$ mice (Fig. 8b) compared with their respective controls. IL-4 potentializes the IL-3 effects on PD-L1 expression up-regulation on basophils ex vivo (Fig. 5b, d) and is required in $Lyn^{-/-}$ mice to enable lupus-like nephritis development as we previously showed[14].

We next assessed whether IL-3 or IL-4 blockade could modulate in vivo the PD-L1 expression on the surface of basophils in the lupus-like context of $Lyn^{-/-}$ mice (Fig. 8c). In line with the above results, IL-3 blockade dampened PD-L1 expression on basophils in spleen and lymph nodes (Fig. 8d and Supplementary Fig. 11b) and reduced overall basophil activation status (CD200R1 expression[21,48]) and accumulation in SLO as IL-4 blockade did (Fig. 8d–f and Supplementary Fig. 11c, d). Blockade of IL-3 or of IL-4 resulted in dampening TFH cell and CD19⁺CD138⁺ cell accumulation in SLO (Fig. 8g, h and Supplementary Fig. 11e, f).

Altogether, these results strongly suggest that the increased IL-3 and IL-4 titers sustain basophil activation and accumulation in SLO in the lupus-like context where they promote their PD-L1 and IL-4-dependent effects on TFH cell and plasmablast accumulations.

### Human basophils drive ex vivo TFH cell and TFH2 cell differentiation through IL-4, IL-6, and PD-1-dependent mechanisms

We next sought to validate the ability of human basophils to promote TFH cell differentiation in a co-culture system. First, CD3- and CD28-activated human naïve CD4⁺ T cells were cultured for three days without or with increasing numbers of purified human basophils demonstrating the capacity of human blood basophils to induce TFH cell differentiation (Fig. 9a–c) more potently than mouse spleen basophils (Fig. 4a). Next, cultures of CD3- and CD28-activated human naïve CD4⁺ T cells alone or together with basophils were repeated in the presence of blocking antibodies targeting IL-4, IL-6, PD-1, or their corresponding isotype controls. As seen in the murine system, IL-4 and PD-1 antagonisms led to a dramatic decrease in the ability of human basophils to drive CD4⁺ T cell differentiation into TFH cells (Fig. 9d). IL-6 blockade also decreased basophil-induced TFH cell differentiation, likely due to its effects on the T cell-derived IL-6 (Fig. 9d). Most of the basophil-induced TFH cells were belonging to the TFH2 cell subset expressing neither CCR6 nor CXCR3 (Fig. 9e, f). This TFH2 cell subset differentiation was dramatically dampened by the blockade of IL-4, IL-6, or PD-1 (Fig. 9e, f) leading mainly to a compensatory TFH1 cell differentiation (Supplementary Fig. 12).

Altogether, these data demonstrate that human basophils could induce human naïve CD4⁺ T cell differentiation into TFH cells, especially into the TFH2 cell subset, and that this was dependent on IL-4 and PD-1. Together with the PD-L1 overexpression by blood basophils from SLE patients (Fig. 1e), these results strongly suggest that, as demonstrated in the lupus-like mouse models (Figs. 2–8), basophils promote the pathogenic accumulation of TFH and TFH2 cells during SLE pathogenesis through their expression of PD-L1 and IL-4. The main findings of the study are summarized in Supplementary Fig. 13.

## Discussion

In this study, we identify mechanisms by which basophils control the pathogenic accumulation of TFH cells in SLO to promote autoreactive IgG production during SLE pathogenesis. Through PD-1/PD-L1

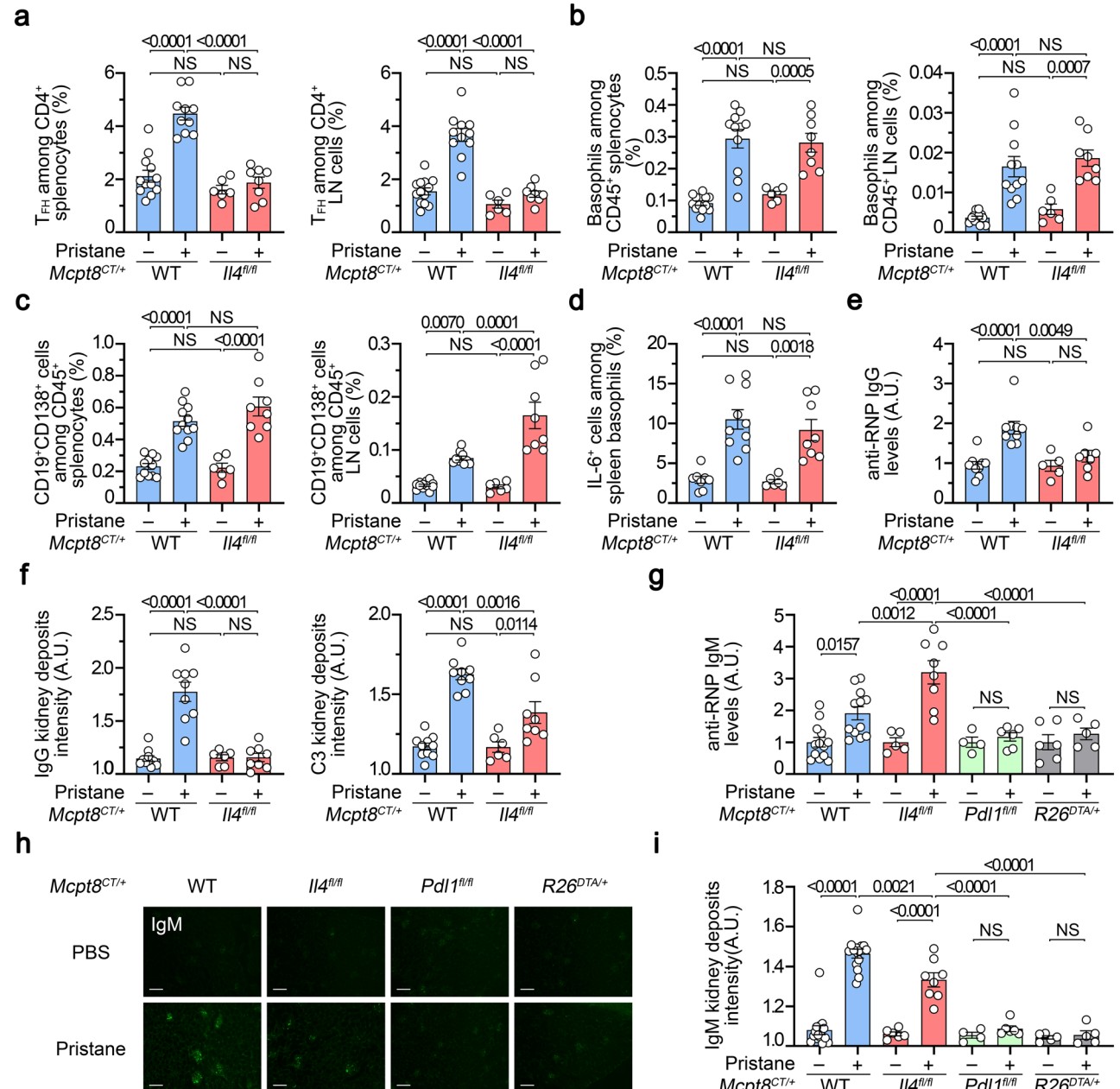

**Fig. 7 | Basophil-derived IL-4 controls T-dependent autoreactive antibody isotype switch in lupus-like disease. a** Proportions (%) of TFH cells among CD4+ T cells in spleen (left) (*n* = 12/10/6/8) and lymph nodes (LN) (right) (*n* = 13/11/6/8) from *Mcpt8CT/+* (WT) (blue) or *Mcpt8CT/+Il4fl/fl* (*Il4fl/fl*) (red) mice injected with PBS (−) or pristane (+). **b** Proportions (%) of basophils among CD45+ cells in the spleen (left) (*n* = 12/11/6/8) and lymph nodes (right) (*n* = 13/11/6/8) from mice as in (**a**). **c** Proportions (%) of CD19+CD138+ cells CD45+ cells in the spleen (left) (*n* = 11/11/6/8) and lymph nodes (right) (*n* = 12/10/6/8) from the mice described in (**a**). **d** Proportions (%) of spontaneous IL-6+ cells among basophils in the spleen from the mice described in (**a**) (*n* = 8/10/6/8). **e** Anti-RNP IgG plasma levels from the same mice as in (**a**) were quantified by ELISA and normalized to the mean of PBS-injected *Mcpt8CT/+* values (*n* = 9/9/5/7). (**f**) Quantification of C3 (left) and IgG (right) glomerular deposits in kidneys from the mice described in (**a**) (*n* = 10/9/6/8). A representative picture for each genotype in each condition is shown in Supplementary Fig. 9c, d. **a–f** The data shown that concerns *Mcpt8CT/+* (WT) mice are the

same as the data shown in Fig. 6. **g** Anti-RNP IgM plasma levels were determined by ELISA and data were normalized to the mean of the PBS-injected control values for each genotype. The mice analyzed were *Mcpt8CT/+* (WT), *Mcpt8CT/+Il4fl/fl* (*Il4fl/fl*), *Mcpt8CT/+Pdl1fl/fl* (*Pdl1fl/fl*) (green) and basophil-deficient (*Mcpt8CT/+R26DTA/+*) (*R26DTA/+*, gray) mice treated with PBS or pristane (− or +, respectively) (*n* = 13/12/5/8/4/6/6/5). **h** Representative pictures of kidneys from mice with the indicated genotypes injected with PBS or pristane showing the intensity of anti-IgM staining by immunofluorescence. Scale bar = 200 μm. **i** Quantification of IgM glomerular deposits in kidneys from the mice described in (**g**) (*n* = 15/15/6/8/4/6/5/5). **a–i** Results are from at least three independent experiments and presented as individual values in bars representing the mean values ± s.e.m. **a–g, i** Statistical analyses were done by two way ANOVA followed by Tukey's multiple comparisons test between the indicated groups. *P* values are shown above each bracket. NS not significant. Source data are provided in the Source Data file.

dependent mechanisms and IL-4 production, basophils controlled TFH cell numbers, cytokine production abilities, TF expressions, and TFH cell functions in the lupus context. Dysregulation of IL-3 titers increased PD-L1 expression by basophils, that was further enhanced by

basophil-derived IL-4, and it supported basophil accumulation in SLO where they acted on TFH cells.

In a normal antigen-driven immunization process, germinal centers (GC) are key structures that allow B cell maturation into

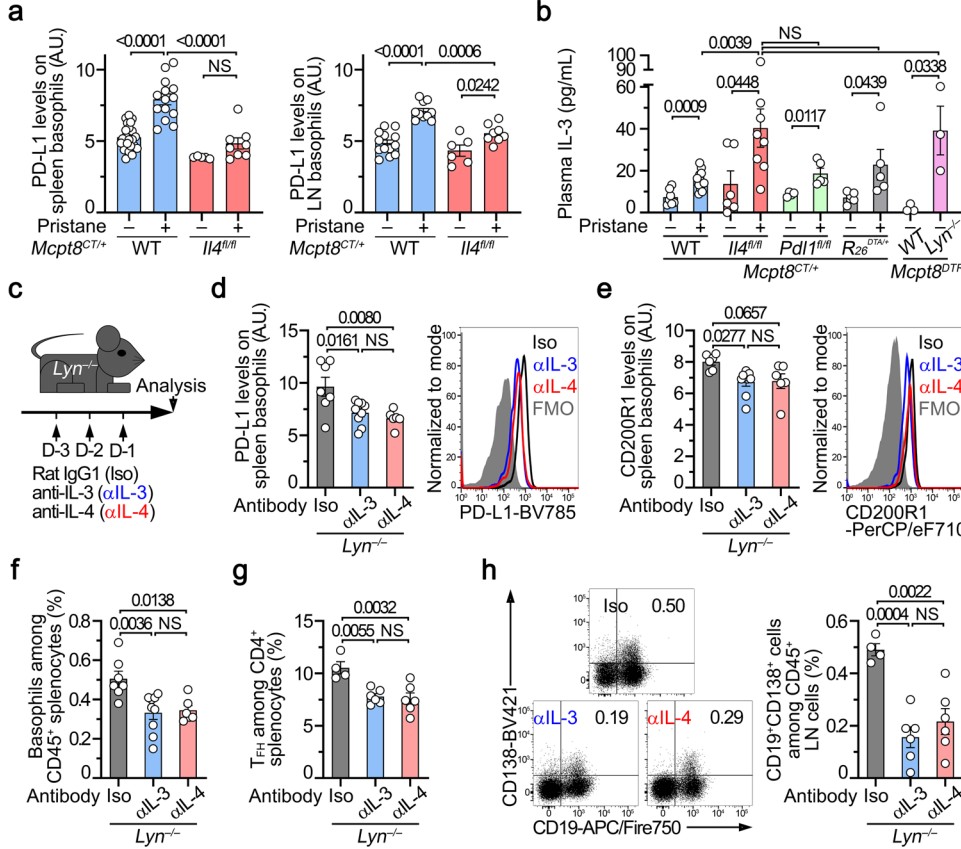

**Fig. 8 | IL-4 and IL-3 control PD-L1 expression, activation and localization of basophils in lupus-like models. a** PD-L1 expression levels on basophils from spleen (left) (n = 22/14/5/8) and lymph nodes (LN) (right) (n = 13/10/6/8) from *Mcpt8^CT/+* (WT) (blue) or *Mcpt8^CT/+Il4^fl/fl* (*Il4^fl/fl*) (red) mice injected with PBS (−) or with pristane (+). **b** IL-3 titers (in pg/mL) in the plasma from *Mcpt8^CT/+* (WT) (blue), *Mcpt8^CT/+Il4^fl/fl* (*Il4^fl/fl*) (red), *Mcpt8^CT/+Pdl1^fl/fl* (*Pdl1^fl/fl*) (green), basophil-deficient (*Mcpt8^CT/+R26^DTA/+*) (gray) mice injected with PBS (−) or with pristane (+) and in aged (min. 30 weeks old) *Mcpt8^DTR* (WT) and *Lyn^−/−Mcpt8^DTR* mice (pink) (n = 7/12/6/8/4/5/6/5/3/3). **c** Aged *Lyn^−/−* mice were injected with isotype control (rat IgG1,κ; Iso; black), rat anti-mouse IL-3 (αIL-3; blue) or rat anti-mouse IL-4 (αIL-4; red) as indicated. PD-L1 (**d**) (n = 7/9/6) and CD200R1 (**e**) (n = 6/9/6) expression levels on basophils from spleen of mice described in (**c**) (*Left*) as measured by flow cytometry

(*Right*). **f** Proportions (%)of basophils among CD45^+ splenocytes of mice as in (**c**) (n = 7/9/6). **g** Proportions (%) of TFH cells among CD4^+ T cells from spleen of mice as in (**c**) (n = 4/6/6). **h** Proportions of CD19^+CD138^+ cells among CD45^+ cells from lymph nodes (LN) (%) of mice as in (**c**) (n = 4/6/6) (*Right*) as measured by flow cytometry (*Left*). **a, b, d–h** Results are from at least two independent experiments and presented as individual values in bars representing the mean values ± s.e.m. **a, b, d–h** Statistical analyses were done by two-way ANOVA test followed by Tukey's multiple comparisons tests (**a**), by one-way ANOVA test followed by two-sided unpaired *t* tests (**b**) between the indicated groups, or by one-way ANOVA test followed by Tukey's multiple comparisons tests (**d–h**) between the indicated groups. *P* values are shown above each bracket. NS not significant. Source data are provided in the Source Data file.

high-affinity and class-switched antibody-secreting cells[49]. The formation and maintenance of these structures depend on TFH cells[7]. Basophils are dispensable to mount an efficient humoral response to OVA protein immunization[30] and do not control basal TFH or OVA-induced TFH cell populations (this study). Importantly, in the latter conditions, basophils do not accumulate in SLO in contrast to the lupus-like context. As previously shown[15], PGD_2 injections to mice allowed to induce accumulation of basophils to SLO. This accumulation occured mainly at the T:B border and led to the increased number of TFH cells and of GC formed following OVA-immunization dependent on basophils. Dysregulated expansion of TFH cells in an SLE-like context occurs through spontaneous GC-like reactions which are favored by an abnormal cytokine milieu and mainly in the extrafollicular (EF) area[7]. Our results indicate that the pathogenic accumulation of TFH cells in SLO in lupus and thus spontaneous TFH cell responses, are fully dependent on SLO-localized basophils (mainly at the T:B border) and more precisely mediated by basophil expression of PD-L1 and IL-4. Moreover, PD-L1 expressing basophils that did not express IL-4 could induce TFH cell- and GC B cell-independent expansion of plasmablasts in lupus models, indicating that basophils, through PD-L1 expression, had a significant impact as well on the EF response.

The impairment of this basophil-dependent pathogenic accumulation of TFH cell in SLO led to a reduction in autoreactive anti-RNP IgG production and IgG kidney deposits, further validating the relevance of both cell types in autoantibody and pathogenic CIC productions in a lupus-like context. Investigating whether basophils control spontaneous EF TFH cell and B cell responses in other autoimmune diseases, such as rheumatoid arthritis or multiple sclerosis, may lead to developing common therapeutic strategies for these different autoimmune conditions that may share some pathophysiological pathways[47].

Recently, Kim et al. showed that TFH2 cells are induced by, and produce some, IL-4. These TFH2 cells are central in humoral autoimmunity, promote autoreactive IgE production with their frequencies increased in both SLE patients and also some SLE-like mouse models[12]. Accordingly, systemic IL-4 blockade dampened TFH2 cell accumulation and their deleterious effects in the *Ets1^ΔCD4* lupus-like mouse model. As exogenous IL-4 induces Gata3 expression in T cells enabling TH2 cell differentiation[50], exogenous IL-4 enabled Gata3^+ TFH2 cell accumulation in an SLE-like context in a similar way[12]. Our data suggest that basophils deliver the required IL-4-induced priming to CD4^+ T cells in a PD-L1-dependent manner that allows TFH and TFH2 cell accumulations in the lupus-like context. Further studies will be

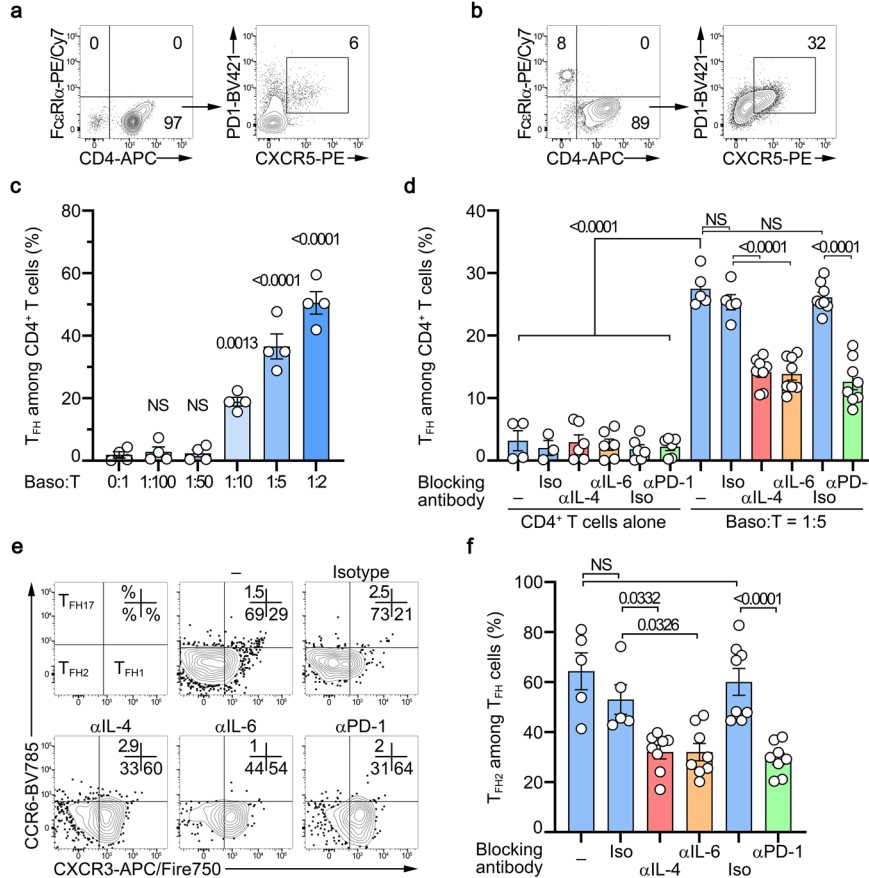

**Fig. 9 | Human basophils drive TFH cell and TFH2 cell differentiation through IL-4, IL-6 and PD-1 dependent mechanisms ex vivo.** Contour plots showing CD3/CD28-activated human CD4+ T cells cultured for three days without (**a**) or with (**b**) purified human basophils at a 5:1 ratio (left) and the condition-induced TFH differentiation of the CD4 + T cells (right). Basophils were defined as FcεRIα+ CRTH2+ CCR3+ cells and TFH cells were defined as CD4+ PD-1+ CXCR5+ ICOS+ cells. **c** Proportions (%) of TFH cells among CD3/CD28-activated CD4+ T cells cultured without (0:1) or with the indicated ratio of purified human basophils (n = 4 per group). **d** Proportions (%) of TFH cells among CD3/CD28-activated CD4+ T cells cultured without or with purified human basophils at a ratio of 1:5 in the absence (−) (blue) or presence of antibodies blocking IL-4 (αIL-4) (red), IL-6 (αIL-6) (orange) or

PD-1 (αPD-1) (green) or the corresponding isotype controls (Iso) (blue) (n = 4/3/6/6/6/5/5/8/8/8). **e** Contour plots showing subsets of TFH cells as defined in Fig. 1 on cells as in (**a**, **b**). **f** Proportions (%) of TFH2 cells among TFH cells after culture as described in (**d**) (n = 5/5/8/8/8). TFH2 cells were defined as CD4+ PD-1+ CXCR5+ ICOS+ CCR6− CXCR3− cells. **c, d, f** Data are presented as individual values in bars representing the mean values ± s.e.m. **c** One representative experiment out of two with cells from 4 different donors is shown. **d, f** Results are from three independent experiments. **c, d, f** Statistical analyses were done by one-way ANOVA test followed by Tukey's multiple comparisons tests between the indicated groups. *P* values are shown above each bracket. NS not significant. Source data are provided in the Source Data file.

required to validate this finding in other lupus-like mouse models including *Ets1^{ΔCD4}* mice bred with basophil-specific PD-L1 deficient mice. This IL-4 mediated basophil-TFH pathogenic axis is important since basophil depletion may represent a safer therapeutic strategy than global IL-4 neutralization. Indeed, this cytokine controls a large number of physiologically protective processes[51,52] and other deleterious pathways involved in SLE pathophysiology may limit the benefits for the patients of long-term IL-4 blockade[2].

Both human and murine basophils promoted CD3/CD28 activated naïve CD4+ T cell differentiation into TFH cells ex vivo in the absence of specific antigen or recognized potent antigen-presenting cells. These effects depended on basophil-expressed IL-4 and PD-L1. PD-1 engagement on CD4+ T cells in the presence of IL-4 could mimic the effects of basophils on TFH cell differentiation by downregulating the repressor Bach2 and inducing some pro-TFH genes. These settings may resume spontaneous TFH cell responses occurring in the SLE context[7]. TCR repertoire analysis of these basophil-induced TFH cells may help to decipher whether these spontaneous TFH cell responses favor autoreactivity of the TFH cell compartment.

pDC and type I IFN have recently been shown to promote EF T-dependent B cell responses to extracellular self DNA[53]. Our data

suggest that basophils and their PD-L1 expression are mandatory to induce TFH cell pathogenic accumulation and B cell response to nuclear antigens. PD-L1 expression by CD11c-expressing cells inhibits TFH cell response in experimental autoimmune encephalomyelitis[54]. Type I IFN promotes human basophil apoptosis, but this effect is rescued by IL-3[55] whose production (mainly by T cells[42]) is increased during SLE pathogenesis[43,46] and acts as well on pDC survival[56]. Thus, it may be relevant to study the interplay between basophils and DC subsets in this process. pDC or follicular DC and basophils may indeed compete to engage PD-1 on TFH cells to respectively inhibit or promote TFH cell expansion and TFH2 cell differentiation in EF responses in an autoimmune context.

PD-L1 expression by basophils has been reported in humans and may be induced by IFNγ in the presence of IL-3[57]. PD-L1 upregulation on human basophils has recently been described during SARS-CoV2 infections[58] although it is not directly induced by the virus[59]. T-cell derived IL-3 and the chemokine CXCL12 are as well upregulated in infected patients with a positive outcome[60]. Interestingly, basopenia occurs in acute SARS-CoV2 infection suggesting putative recruitment of basophils to SLO[61]. This hypothesis is further supported by basophil overexpression of CXCR4 and CD62L in the SARS-CoV-2 infection

setting, similar to what we described in SLE patients[14,15,61,62]. Normalization of both blood basophil numbers and activation markers is associated with the humoral response and recovery of COVID-19 patients[61,63]. This may indicate a role for PD-L1-overexpressing basophils in TFH cell induction to promote antiviral humoral response. EF responses occurring in lupus-prone mice and during some infections can quickly involute if new plasmablasts are not generated[49]. Thus, controlling basophil function or localization in SLO during viral infections may represent interesting therapeutic strategies to promote or sustain the antiviral humoral response.

Unexpectedly, when injected with pristane, mice with IL-4-deficient basophils still showed some plasmablast accumulation in SLO and C3 kidney deposits despite the absence of TFH cell accumulation, anti-RNP IgG induction, and IgG kidney deposits. Unlike pristane-injected basophil-deficient mice or mice with PD-L1-deficient basophils, $Mcpt8^{CT/+} Il4^{fl/fl}$ mice developed high titers of anti-RNP IgM. This indicates that basophils express factors other than IL-4 in SLO during lupus-like disease responsible for their PD-L1-dependent action on B cell maturation into plasmablast in the absence of TFH cell and GC B cell pathogenic accumulations. However, it strongly suggests as well that basophil-derived IL-4 is mandatory for autoantibody class switch towards IgG in the lupus-like context. Further characterization of this B cell–basophil relationship in SLO in the SLE context will be required to identify basophil-derived factors that contribute to TFH-independent plasmablast proliferation.

Three main points remain to be further elucidated following our present study. First, the effects of basophil-specific PD-L1 deficiency should be validated in more severe and more classical lupus-like mouse models such as NZBxNZW F1 or MRL-$Fas^{lpr}$. Second, basophil positioning and dynamic in SLO as well as characterization of molecular in situ TFH cell-basophil interactors (including PD-1/PD-L1) need to be described in SLE patient SLO samples and further detailed in lupus-like models. A combined approach by spatial transcriptomics and proteomics would give a comprehensive view of basophil positioning and cellular interactions in SLO samples. Third, how basophils and B cells interact through basophil-expressed PD-L1 will be a stimulating study. Basophils may influence the EF response in the lupus context by interacting with PD-1 expressing B cells[64].

Our study establishes a direct link between basophils and TFH cells in the SLE context that promotes autoreactive IgG production and lupus nephritis pathogenesis. Altering the basophil/TFH cell axis in the SLE context may thus represent a promising therapeutic strategy in SLE. As specific antibody-mediated basophil depletion in humans is not yet available, acting on basophil activation through IL-3 blockade[43,46], anti-IgEapproach[23,65], or preventing basophil accumulation in SLO through PTGDRs antagonisms[15,21] represent a promising area of clinical development that may provide to SLE patients some new specific and efficient therapeutic options.

## Methods

### Patient recruitment and ethics
Blood samples were collected from adult patients enrolled in a prospective long-term study of SLE and chronic renal diseases from April 2015 to February 2020. All SLE patients fulfilled the American College of Rheumatology (ACR) classification criteria for SLE. SLE and healthy control (CT) donor characteristics are shown in Supplementary Table 1. SELENA-SLEDAI (Safety of Estrogens in Lupus Erythematosus National Assessment - SLE Disease Activity Index) scores were assessed to evaluate patients' lupus activity who were classified as inactive (0–1), mildly active termed "mild" (2–4), and moderately to severely active termed "active" (>4). Pregnant and/or HIV, HBV and HCV seropositive patients were excluded from the study to avoid any interference in the observed immunological phenotypes. The study and the use of human material have been approved by the institutional ethics review committee "Comité Régional de Protection des

Personnes" (CRPP, Paris, France) in April 2014 under the reference ID-RCB 2014-A00809-38. SLE samples were obtained from in- and outpatients and clinical data were harvested after approval by the Commission Nationale de l'Informatique et des Libertés (CNIL). Healthy controls were recruited among volunteering healthcare workers and through the Etablissement Français du Sang (EFS). All samples were collected in heparinized tubes (BD vacutaine) and processed within 4 h. Written informed consent was obtained from all individuals.

### Human sample handling
Heparinized human blood samples were centrifuged at 600 g for 5 min and 2 mL of plasma were collected and stored at −80 °C for later analysis. Red blood cells (RBC) were lysed in RBC lysing buffer (150 mM $NH_4Cl$, 12 mM $NaHCO_3$, 1 mM EDTA, pH 7.4) in a ratio of 5 mL of blood for 20 mL of ACK lysing buffer. After 5 min of incubation at room temperature (rt), 25 mL of PBS were added and cells were centrifuged at 600 g for 5 min, and the supernatant was discarded. This procedure was repeated 3 times. Leukocytes were then resuspended in fluorescence-activated cell sorting (FACS) buffer (PBS 1% BSA, 0.01% $NaN_3$, 1 mM EDTA) and prepared for flow cytometry (see below). Leukocyte count and viability (>95%) were assessed by trypan-blue staining on a Malassez hemacytometer.

### Mice
$Mcpt8^{DTR66}$, $Lyn^{-/-} Mcpt8^{DTR15,67}$, $Il4^{fl/fl35}$, $Il6^{fl/fl36}$, and $Pdl1^{fl/fl37}$ mice were on a pure C57BL/6J background and bred in our animal facilities ($Lyn^{-/-}$: JAX stock# 003515). Rosa26-loxP-Stop-loxP-DTA C57BL/6J (B6.129P2-Gt(ROSA)26$^{Sortm1(DTA)Lky/J}$; JAX stock# 009669) ($R_{26}^{DTA/DTA}$ or $R_{26}^{DTA/+}$) mice[68] were purchased from The Jackson Laboratory through Charles River Laboratories. CT-M8 ($Mcpt8^{tm1.1(cre)lcs}$ or $Mcpt8^{CT/CT}$ or $Mcpt8^{CT/+}$) mice were recently described[16]. The mice crossed in our animal facilities $Mcpt8^{CT/+}$; $Mcpt8^{CT/+} Il4^{fl/fl}$; $Mcpt8^{CT/+} Il6^{fl/fl}$; $Mcpt8^{CT/+} Pdl1^{fl/fl}$ and $Mcpt8^{CT/+} R_{26}^{DTA/+}$ were on a C57BL6J/N mixed genetic background at the F2 generation. For lupus-like disease analysis of the $Lyn^{-/-}$ model, "aged" $Mcpt8^{DTR}$ and $Lyn^{-/-} Mcpt8^{DTR}$ age-matched and sex-matched mice were analyzed between 30 and 45 weeks of age (50% males and 50% females). For lupus-like disease analysis of the pristane-induced model, only female mice were analyzed as described below as a sex bias, as in human patients, is observed in this model[69]. Mice were maintained under specific pathogen-free conditions in our animal facilities with access to tap water and chow ad libidum, 12 h dark/light cycles, an ambient temperature kept between 22 °C and 24 °C and a 55 ± 10% hygrometry. All mice were euthanized by $CO_2$ inhalation in a regulated chamber (TemSega, France). The study was conducted in accordance with the French and European guidelines and approved by the local ethics committee comité d'éthique Paris Nord N°121 and the Ministère de l'enseignement supérieur, de la recherche et de l'innovation under the authorization number APAFIS#14115.

### Pristane-induced lupus-like mouse model
Pristane-induced lupus-like disease was initiated by injecting 500 μL of 2,6,10,14-tetramethyl-pentadecane or Pristane (Sigma) into the peritoneal cavity of 7–10 weeks-old female mice. For control individuals, genotype- and age-matched female mice were injected with 500 μL of phosphate buffer saline pH 7.4 (PBS) (Gibco). Mice were attributed to PBS- or pristane-injected groups randomly and maintained in the same cage for the whole procedure. Pristane- and PBS-injected mice were analyzed 8 weeks or 24 weeks after injection.

### DT-mediated basophil depletion in lupus-like context
For diphtheria toxin (DT)-mediated basophil depletion, $Mcpt8^{DTR}$, and $Lyn^{-/-} Mcpt8^{DTR}$ mice were injected intraperitoneally with 100 μL of PBS containing (or not) 1 μg of DT (Sigma) 10, 9, 6, 2 and 1 day before euthanasia.

## Mouse OVA immunization experiments

10 to 15 weeks old C57/BL6J *Mcpt8^DTR* or *Lyn^−/−xMcpt8^DTR* sex-matched mice were immunized by intraperitoneal injection of 200 μL of a 50/50 emulsion of Alum (Thermofisher Scientific) with 100 μg of ovalbumin (OVA, Sigma-Aldrich) diluted in PBS or with PBS alone for control mice. A similar injection was performed on day 7 and the mice were analyzed on day 14. For DT-mediated basophil depletion, 1 μg of DT (or PBS as a control) was injected intraperitoneally on day 12 and day 13. To induce basophil accumulation in spleen and draining lymph node (mesenteric lymph node, mLN) during the immunization procedure, on day −1, 3, 6, 9, 12, and 13, the concerned mice were injected intraperitoneally with 20 nmoles of prostaglandin D$_2$ (Cayman Chemicals) in 100 μL of PBS per injection (Supplementary Fig. 4)[15].

## Anti-IL-3 and anti-IL-4 antibodies treatment

Aged *Lyn^−/−* mice (min. 25 weeks old) were injected intraperitoneally with 100 μg of isotype control (rat IgG1,κ; clone HRPN, BioXcell), 50 μg[43] of rat anti-mouse IL-3 (clone MP2-8F8, Biolegend) or 100 μg[70] of rat anti-mouse IL-4 (clone 11B11, Biolegend) 72, 48 and 24 h before analysis.

## Mouse sample processing

Mice were euthanized in a controlled CO$_2$ chamber (TEM Sega) and blood sampling was performed through cardiac puncture with a heparin-coated syringe with a 25 G needle. Blood was centrifuged at 300 g for 15 min and plasma was harvested and kept at −80 °C for later analysis. RBC were lysed in 5 mL of RBC lysing buffer for 5 min at rt and washed with 10 mL of PBS. This procedure was repeated 3 times and cells were resuspended in PBS. The left kidney was harvested and embedded in OCT embedding matrix (Cellpath) and snap-frozen in liquid nitrogen and kept at −80 °C for later analysis. For cell analysis, spleen and lymph nodes (cervical, brachial, and inguinal) were harvested in PBS and dissociated by mechanical disruption on a 40 μm cell strainer (Falcon, Corning). For splenocytes, RBC were lysed once in 5 mL RBC lysing buffer 5 min at rt and washed with 10 mL of PBS. Cell counts were assessed by trypan-blue staining on a Malassez hemacytometer and 1 to 5 million cells were used per FACS staining condition. For immunofluorescence analysis, spleen was harvested and embedded in OCT embedding matrix (Cellpath) and snap-frozen in liquid nitrogen, and kept at −80 °C for later analysis.

## Ex vivo stimulation of splenocytes

Mouse splenocytes were harvested as described above and resuspended at 5 million cells/mL in culture medium (RPMI 1640 with Glutamax and 20 mM HEPES, 1 mM Na-pyruvate, non-essential amino acids 1X (all from Life Technologies), 100 μg/mL streptomycin and 100 μ/mL penicillin (GE Healthcare) and 37.5 μM β-mercaptoethanol (Sigma-Aldrich) supplemented with 20% heat-inactivated fetal calf serum (FCS) (Life Technologies)). For phorbol-myristate-acetate (PMA) and ionomycin stimulation experiments, whole splenocytes were stimulated or not with 40 nM of PMA and 800 nM ionomycin for 4 h in the presence of 2 μg/mL of brefeldin A (all from Sigma Aldrich, Merck) and cultured at 37 °C and 5% CO$_2$. For IL-3, IL-4 or anti-IgE stimulations, cells were stimulated with the doses indicated in the figure legends for 2 or 20 h at 37 °C and 5% CO$_2$. Then, cells were harvested by repeated flushing, and wells were washed with 1 mL of PBS. Samples were then prepared for flow cytometry analysis.

## Flow cytometry staining

For human leukocytes, non-specific antibody binding sites were saturated with 20 μL of a solution containing 100 μg/mL of human, mouse, rat, and goat IgG (Jackson ImmunoResearch Europe and Innovative Research Inc) in FACS buffer. 200 μL of staining solution containing the panel of fluorophore-conjugated specific antibodies or their fluorophore-conjugated isotypes (described in Supplementary Table 3) were added to the cells for 30 min at 4 °C protected from light.

After a wash in PBS, cells were fixed in fixation buffer (Biolegend) for 20 min at 4 °C and then washed in FACS buffer before data acquisition. For mouse samples, cells washed in PBS were stained with GHOST 510 viability dye (TONBO) following the manufacturer's instructions. Non-specific antibody binding sites were saturated with 10 μg/mL of anti-CD16/CD32 antibody clone 2.4G2 (BioXCell), and 100 μg/mL of polyclonal rat, mouse, and Armenian Hamster IgG (Innovative Research Inc.) in FACS buffer and stained with the antibodies described in Supplementary Table 3 for 30 min in the dark at 4 °C. Cells were then washed in FACS Buffer before data acquisition. For intracellular staining, cells were first stained extracellularly as described above. Cells were washed in PBS and fixed with fixation buffer for 20 min at 4 °C. Cell permeabilization and intracellular staining were realized with permeabilization/wash buffer (Biolegend) following the manufacturer's instructions. Cells were then resuspended in FACS buffer before acquisition All flow cytometry acquisitions were realized using a Becton Dickinson 5 lasers LSR Fortessa X-20 and data analysis using Flowjo vX (Treestar, BD Biosciences). For assessment of surface marker expression levels, ratios of the geometric mean fluorescence intensity (gMFI) of the markers to the gMFI of the corresponding isotype control were calculated and expressed in arbitrary units (A.U.).

## Kidney Immunofluorescence assays

4 μm thick cryosections of OCT-embedded kidneys were fixed 20 min in ice-cold acetone and kept a −80 °C until immunofluorescence staining. Slides were thawed and fixed in 10% formalin (Sigma) for 20 min at room temperature (rt) and blocked with PBS containing 1% BSA (Euromedex) and 5% goat serum (Sigma-Aldrich). Slices were stained for 2 h at room temperature in the dark in a humid chamber with FITC-conjugated anti-mouse C3 (Cedarlane), Alexa Fluor® 488-conjugated anti-mouse C3 (Santa Cruz Biotech), Alexa Fluor® 488-conjugated goat anti-mouse IgG Fcγ-specific (Jackson Immunoresearch), FITC-conjugated goat anti-mouse IgM (BioRad (AbD Serotec)) or corresponding isotype controls. Slides were mounted in Immunomount (Thermofischer Scientifics) and analyzed by fluorescent microscopy (Leica DMR, Leica microsystems). The ratio of specific glomerular fluorescence over tubulointerstitial background was then measured using ImageJ software v. 1.43 u (NIH), averaging at least 30 glomeruli per mouse for each sample.

## Spleen immunofluorescence assays

5 μm thick cryosections of OCT-embedded spleens were fixed 20 min in ice-cold acetone and kept a −80 °C until immunofluorescence staining. Slides were thawed and fixed in 4% paraformaldehyde for 20 min at rt, washed 5 times in washing buffer (WaB) (PBS, 10% FCS, 0.05% Triton X-100) and incubated 10 min at rt in 50 mM NH$_4$Cl. Slides were then treated as follows with 5 washes between each step: Saturation as described for flow cytometry in WaB for 1 h at rt; avidin/biotin blocked (2 × 15 min) (R&D systems); incubation 2 h with 2 μg/mL homemade biotinylated anti-mMCP8 (clone TUG8 and Sulfo-NHS-LC-Biotine EZ-Link kit) in WaB. Slides were then incubated o/n at 4 °C with the following conjugated-reagents: 20 μg/mL of BV421-anti-CD4 (clone RM4-5), AF647-GL7, AF488-anti-B220 (clone RA3-6B2) or FITC-anti-IgE (clone RME-1) and 1 μg/mL of AF594-strepatividin. After washes in WaB and then PBS, slides were mounted with Epredia Immu-Mount reagent (Thermo Scientific). Within 3 h, images were acquired with a 4 lasers LSM 780 Zeiss confocal microscope with a x10 objective using Zen 2.1 software (v11.0.4.19) and with a 4 lasers Leica sp8 confocal microscope with a x63 objective using LAS X software (v3.5.5.19976). Image processing was done using Image J software (v1.54 f) and quantifications were done with QuPath software (v. 0.5)[71].

## Anti-RNP IgG and IgM autoantibody detection

Maxisorp 96 well plates (Thermo Scientific) were coated overnight at 4 °C with 10 μg/mL of purified RNP complexes (Immunovision) diluted in carbonate buffer (100 mM NaHCO$_3$ and 30 mM Na$_2$CO$_3$ pH 9.6).

Plates were washed 3 times in PBS containing 0.1% of Tween-20 (Bio-Rad laboratories) (PBS-T) and saturated for 1 h with PBS containing 5% of FCS. For anti-RNP IgG quantification, plasma samples were diluted 1:25 in PBS-T containing 5% of goat serum and 100 µL added to the wells. Samples, positive and negative controls were incubated for 2 h at rt. Plates were washed 5 times with PBS-T. 100 µL of either 500 ng/mL goat anti-mouse IgG (Invitrogen) or 10 ng/mL goat anti-mouse IgM (Bethyl laboratories) conjugated to horseradish peroxidase (HRP) diluted in PBS-T containing 5% goat serum were added and incubated 1 h at room temperature. After 5 washes in PBS-T, 100 µL of tetramethylbenzidine (TMB) substrate (ThermoFisher) were added to the wells and incubated at least 20 min at rt and the reaction was stopped with 0.2 N sulfuric acid solution. Optical density at 450 nm was measured by spectrophotometry (Infinite 200 Pro plate reader, TECAN). On each plate, similar negative and positive controls were run. Optical density (OD) values were first normalized to the negative controls and then, the presented results were normalized to the mean of the control mice values and expressed in arbitrary units (A.U.).

### Anti-dsDNA IgG detection
Maxisorp 96 well plates (Thermo Scientific) were coated overnight at 4 °C with calf thymus dsDNA (Sigma-Aldrich) diluted in TE buffer (Tris-HCl 10 mM, EDTA 1 mM pH 9) at a concentration of 2 µg/mL and diluted in the same volume of Pierce DNA coating solution (Thermo Scientific) to obtain a final concentration of 1 µg/mL of dsDNA. The same protocol as for anti-RNP IgG was then followed but the PBS-T contained 0.05% of Tween-20.

### Cell sorting and co-culture
For basophil/CD4[+] T cell mouse co-culture experiments, F(ab')2 anti-CD3 (clone145-2C11 at 0,5 µg/mL) and anti-CD28 (clone PV-1 at 0,5 µg/mL) antibodies (BioXcell) were coated overnight at 4 °C in culture 96 well plates (Costar) in sterile-filtered coating buffer (20 mM carbonate buffer (pH 9.6) containing 2 mM MgCl$_2$ and 0.01% NaN$_3$). Wells were washed in PBS before plating the cells. Spleens were harvested and handled as described above and resuspended at $10^8$ cells/mL in PBS containing 2% FCS and 2 mM EDTA. Naïve CD4[+] T cells were sorted by magnetic negative selection following manufacturer protocol (Stemcell Technologies) and resuspended in culture medium at 0.5 million cells/mL. 100 µL of naïve CD4[+] T cell suspension were then added to wells of interest. Basophils from CTM8 mice were purified over 98% by electronic sorting using the BD FACSMelody Cell Sorter (BD Biosciences) using the tandem tomato fluorescent protein as a specific marker[16]. Basophils were re-suspended in culture medium at 0.1 million cells/mL and 100 µL were added to wells of interest. All wells were supplemented with 10 pg/mL of recombinant mouse IL-3 (Peprotech). After 3 days of culture at 37 °C 5% CO$_2$, cells were harvested and prepared for flow cytometry as described above. For cytokine production detection, cells were stimulated or not for the last 4 h with 40 nM PMA and 800 nM ionomycin in the presence of 2 µg/mL brefeldin A (Sigma-Aldrich, Merck) and prepared for flow cytometry as described above.

For TFH cells isolation, spleen CD4[+] T cells were enriched from $Mcpt8^{DTR}$ WT and $Lyn^{-/-}$ mice basophil-depleted or not (+/- DT) using EasySep mouse CD4[+] T cell isolation kit (Stemcell technologies) following the manufacturer's instructions. Cells were then stained with anti-CD45 PE/Cy7, anti-CD4 APC/Fire750, anti-CD25 AF488, anti-CD44 PE/Dazzle594 and biotin anti-CXCR5 antibodies for 30 min at 4 °C. After washing, cells were incubated with AF647 streptavidin for 20 min at 4 °C. TFH cells were FACS-sorted as CD45[+] CD4[+] CD25[−] CD44[hi] CXCR5[+] and purity was checked after each sort (>93% of CD45[+] CD4[+] CD25[−] CD44[hi] CXCR5[+] PD-1[+] TFH cells). $5 \times 10^4$ sorted TFH cells were co-cultured for 3 days in culture medium with $5 \times 10^4$ B cells isolated from WT splenocytes using EasySep mouse Pan-B cell isolation kit (Stemcell

technologies) in anti-CD3/anti-CD28 coated plates as described above. After 3 days, cells were harvested and stained with anti-CD45 PE/Cy7, anti-CD4 APC/Fire750, anti-CD138 BV421, and anti-CD19 PE/Dazzle594 antibodies and plasmablasts among CD45[+] cultured cells were defined as CD4[−] CD19[+] CD138[hi] cells.

For human co-culture experiments, 96 well plates (Costar) were coated overnight at 4 °C in coating buffer containing 5 µg/mL of mouse anti-human CD3 (clone OKT3) (Thermo Fischer Scientific). Blood from healthy volunteers was handled and lysed as described above in sterile conditions and re-suspended at $5 \times 10^7$ cells/mL in PBS 2% FCS (Gibco) 2 mM EDTA. Naïve CD4 + T cells and basophils were sorted by magnetic negative selection following manufacturer instructions (Stemcell Technologies). Naïve CD4[+] T cells were re-suspended in culture medium at 0.5 million cells/mL. and 100 µL of cell suspension were then added to each well. Basophils were resuspended in culture medium at 0.1 million cells/mL and 100 µL were added to each well unless otherwise specified. All wells were supplemented with 10 pg/ml of recombinant human IL-3 (Biolegend) and 2 µg/mL of mouse anti-human CD28 antibody (Clone CD28.2 at (BioXcell)). When indicated, some wells were supplemented with mouse anti-human blocking antibodies targeting IL4 (Clone MP4-25D2), IL-6 (Clone MQ2-13A5), both at 5 µg/mL or PD-1 (Clone EH12.2H7 at 10 µg/mL) or with the corresponding isotypes at the same concentrations. Cells were harvested after 3 days of culture at 37 °C 5% CO$_2$ and prepared for flow cytometry as described above.

### Reverse transcription and quantitative polymerase chain reaction (RT-qPCR)
RNA extraction on re-sorted cells was performed as described in the manufacturer's protocol (RNeasy Mini kit, Qiagen). cDNA was synthesized with iScript cDNA Synthesis Kit (Bio-Rad). Quantitative PCR was performed with SsoAdvanced SYBR green reaction mix (Bio-Rad) using the following KiCqStart™ Primers pairs (purchased from Sigma-Aldrich): M_Actb_1; M_Batf_1; M_Bcl6_1; M_Gata3_1; M_Maf_1, M_Prdm1_1, and M_Cxcr5_1 to respectively measure β-actin, Batf, Bcl6, Gata3, Maf, Prdm1, and Cxcr5 mRNA expression levels. Bach2 primers were purchased from Integrated DNA Technology. All sequences are reported in supplementary Table 3. Quantitative PCR was performed on the CFX96 Touch Real Time PCR Detection System (Bio-Rad) and following amplification, Ct values were obtained using the CFX Manager™ software 3.1 (Bio-Rad).

### Statistical analysis
All tests used in this study were two-sided. We determined normal distribution in each group by using d'Agostino-Pearson omnibus normality test. If the distribution was gaussian, or if the sample size (n) was less than 8, we used Student's unpaired $t$ tests to compare the differences of one variable between two groups. For non-parametric distributions, we used Mann–Whitney U tests. When comparing more than two groups, one-way analysis of variance (ANOVA) or Kruskal-Wallis tests with multiple comparison post-tests, as specified in the figure legends, based on the distribution of values in each group, were used. When comparing two variables in more than two groups, two-way analysis of variance (ANOVA) test with Tukey's multiple comparison post-tests were used. Covariations were calculated using Spearman's rank correlation coefficient (r) as the distributions of the variables were not gaussian. When $p < 0.05$, the analyzed data pairs were considered positively correlated when $r > 0.2$ and negatively correlated when $r < -0.2$ with the slope of the corresponding linear regression analysis significantly different from 0. A r absolute value between 0.3 and 0.5 was considered as "fairly correlated", and above 0.5 as "strongly correlated". In all figures, where $p < 0.0001$ is indicated, the p value was too low for Prism software to provide an exact value. All statistical analyses were performed using Prism v9.4-v10.2 software (Graphpad).

## Material availability

Further information and requests for resources and reagents should be directed to and will be fulfilled by the corresponding author, Nicolas CHARLES (nicolas.charles@inserm.fr). This study did not generate new unique reagents. The CT-M8 mouse line Mcpt8$^{tm1.1(cre)lcs}$ (Accession ID: MGI:7327249) will be made available upon a Material Transfer Agreement fulfillment. All reagents and resources used in this study are listed in Supplementary Table 3.

## Reporting summary

Further information on research design is available in the Nature Portfolio Reporting Summary linked to this article.

# Data availability

All data reported in this paper will be shared by the corresponding author upon reasonable request. Source data are provided with this paper.

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

## Acknowledgements

We acknowledge the expert work from the members of the animal core facility (I. Renault, S. Olivré, A. Bouhalfaia), the flow cytometry core facility (G. Gautier, J. Da Silva and V. Gratio) and the imaging facility (S. Benadda) of the Centre de Recherche sur l'Inflammation (INSERM UMR1149), and the help from O. Thibaudeau and L. Wingertsmann from the morphology core facility (INSERM UMR1152). This work was supported by the Institut National de la Santé et de la Recherche Médicale (INSERM), by the Centre National de la Recherche Scientifique (CNRS), by Université Paris-Cité and by the following grants: Fondation pour la Recherche Médicale (FRM), grant # EQU201903007794 to N.C., Agence Nationale de la Recherche (ANR), grants # ANR-19-CE17-0029 BALUMET to N.C. and #ANRPIA-10-LABX-0017 INFLAMEX to U.B. and N.C., Ministerio de Economía y Competitividad y Fondo Europeo de Desarrollo Regional, grant # RTI2018-101105-B-I00 to J.H.

## Author contributions

J.T. designed experiments, conducted experiments, analyzed the data, and wrote the manuscript. Q.S. designed experiments, conducted experiments, analyzed the data, and edited the manuscript. N.C. conceived the project, designed experiments, conducted experiments, wrote the manuscript, and directed the project. L.C., M.K.T., S.V., L.S., Y.L., F.S., E.P., J.B-C, C.P., T.P., U.B., M.B., G.H., K.S., and E.D. conducted experiments, analyzed the data, and/or edited the manuscript. H.K. and K.M. provided the *Il4*^fl/fl^ and *Mcpt8*^DTR^ mice and edited the manuscript. J.H. provided the *Il6*^fl/fl^ mice and edited the manuscript. P.G.F. provided the *Pdl1*^fl/fl^ mice and edited the manuscript. N.C. had full access to all of the data in the study and take responsibility for the integrity of the data and the accuracy of the data analysis. All authors approved the final version of the article.

## Competing interests

N.C. holds a patent related to compositions and methods for treating or preventing lupus (WO20120710042). C.P. and N.C. are coinventors of the patent WO2016128565A1 related to the use of PTGDR-1 and PTGDR-2 antagonists for the prevention or treatment of systemic lupus erythematosus. No other disclosures relevant to this article are reported. The remaining authors declare no competing interests relevant to this article.
