## [Peer Review File · Nature Communications]

PD-L1- and IL-4-expressing basophils promote pathogenic accumulation of T follicular helper cells in lupusREVIEWER COMMENTS

Reviewer #1 (expert in basophils):

INTRODUCTION

- "...basopenia that reflects their accumulation into SLO...". The finding of basopenia has been recently confirmed in pediatric SLE as well (see: *Diagnostics (Basel)*. 2022 Jul 12;12(7):1701. doi: 10.3390/diagnostics12071701; and, *Autoimmun Rev*. 2021 Apr;20(4):102790. doi: 10.1016/j.autrev.2021.102790). In detail, the first study above found some association with disease activity (SLEDAI), but not lupus nephritis. One point of the discussion was related to the sampling time point (in light of the ongoing therapy and previous disease course), which could be relevant also for the study participants enrolled in the present study.
- "In this context, targeting the addressing of basophils to SLO has demonstrated promising therapeutic potential." The authors could clarify the therapeutic tools which demonstrated this potential. In this regard, the authors could also mention the experience with omalizumab, which also alters basophil homeostasis by targeting IgE receptors (see: *Arthritis Rheumatol*. 2019 Jul;71(7):1135-1140. doi: 10.1002/art.40828)
- Other than these minor points, the introduction is clear and comprehensive.

METHODS

- "Blood samples were collected from adult patients enrolled in a prospective long-term study of systemic lupus erythematosus (SLE) and chronic renal diseases." I would ask the authors to specify the study period.
- A specific ethical statement section could be created, including all the IRB specifications that the authors included in the mice section. Actually, specifications about approval for human and mice studies should be clarified. As regards the IRB approval, the approval date should be also stated. In this section, clarifications about the study participants' informed consent should be also included.
- I would suggest to clearly separate the methods related to mice experiments and human analyses.
- statistical methods: "When distribution were Gaussian or when N was too small,..." Please, can you clarify when N is considered too small? Also, can you correct the grammar inconsistency?
- statistical methods: "coupled with ad hoc multiple comparisons post-tests, as indicated in the figure legends depending on the distribution of the values in each group,..." Please, can you better clarify why different post hoc tests were used and, then, the criteria for this differential application?
- For FACS analysis, especially in humans, can you clarify if these were performed on fresh samples or whether the biological samples were stored and analyzed in batch? In the latter case, can you provide the details?
- As regards patients, can you clarify if specific inclusion and exclusion criteria were adopted to enroll them as study participants?
- "...patients' lupus activity who were classified as inactive (0-1), mild (2-4), and active (> 4)." I may suggest classifying patients as mildly active and moderately-severely active or anyway to revise this terminology. Then, it should be corrected in the figures and in the text accordingly.

RESULTS

- "We and others reported that peripheral basopenia and activation of blood basophils correlate with disease activity in SLE patients". Notably, this association was found in the only study assessing basophils in pediatric SLE patients, which should be mentioned as well.
- Following the last comment about the methods section, it would be important to clarify some clinical and demographic aspects of the patients included in this study, as well as the timing of the blood sampling, especially based on the clinical course and the ongoing/previous therapies. This information (which is included in the supplementary table 1) could be summarized in the first part of the results.
- In supplementary table 2, there is a column describing healthy controls. This controls should be defined in the methods section as well. I mean, who are they? How were they recruited?
- supplementary figure 1: panel F shows a covariation analysis and, in the caption, is reported as performed by Spearman coefficient. However, in the statistical methods, covariation analysis is not described at all. Please, complete the methods accordingly, also by defining the levels of correlation and statistical significance.
- In supplementary figure 1, in several panels among b-e, the statistical significance between mild and inactive does not seem to be represented: can you clarify?
- A similar clarification like the previous comment could be needed for figure 1. Following some previous comments about the classification of disease activity, one may guess that differences between all active SLE patients (mild + active) and inactive SLE patients could be not significant. Is this correct?
- Is it possible to clarify in the text and/or figures how many patients were included in each group?
- Similarly, as regards all murine experiment, I would suggest the authors to clarify the number of mice for each group, even if this could be probably understood from the figures itself. However, it would be more immediate if the authors disclose how many mice they used for each experiment/group.
- Beyond these minor aspects, I think that the results are explained well and with a clear logical flow.

DISCUSSION

- Are there any study limitations in the experimental model or, in general, in the study protocol, which may leave some doubts about the conclusions and main findings? If so, please, discuss and explain them.
- In this regard, some specific discussion about the differences (and, then, limitation in the translational potential) between human SLE and murine model could be also included.
- At the beginning of the discussion, I would suggest the authors to schematically list the main and novel findings emerging from this study.
- A very recent paper investigating the role of basophils and specific role of autoreactive IgE may further support and enrich the discussion (see: *Rheumatology (Oxford)*. 2023 Feb 22:kead082. doi: 10.1093/rheumatology/kead082).

Reviewer #2 (expert in TFH cells):

In Tchen et al the authors are assessing the role of Basophils in controlling pathogenic Tfh responses during lupus-like disease. The authors find that Basophils in SLE patients express higher levels of PD-L1 compared to control patients. Utilizing pristane or Lyn-/- models of autoimmunity, the authors find that

basophils promote Tfh and IL-21 responses. In addition, the authors find that basophil PD-L1 has roles during basophil-mediated Tfh promotion. Overall, understanding the signals that promote pathogenic Tfh responses in SLE is an important topic. However, this study lacks sufficient conceptual advance and mechanistic insights to be impactful. In particular, the authors have already demonstrated that basophils control SLE-like disease in vivo and the phenotype on Tfh cells is predictable since these cells promote autoantibodies. Moreover, attempts at uncovering mechanisms are performed utilizing in vitro assays that do not recapitulate Tfh biology. More specific in vivo experiments need to be utilized to elucidate how PD-L1 on basophils may have roles in controlling pathogenic Tfh cells and SLE-like disease.

Major points

1. The authors have already shown in multiple publications that basophils promote autoreactive antibody and plasmablasts in pristane/Lyn-/- autoimmunity models. The alteration in Tfh cells may be an indirect consequence of loss of pathogenic B cells and not necessarily due to an interaction with basophils. No mechanistic studies are performed in vivo to understand the nature of the basophil/Tfh interaction nor at which stages Tfh cells are altered. More sophisticated in vivo studies along with transcriptomics techniques are required to explore this in more detail.
2. The authors suggest the increase in Tfh cells upon basophil deletion is pathogenic because of small changes in IL-21 by flow cytometry, yet it is unclear if these Tfh cells are truly participating in autoimmunity progression. Basophils may be promoting CXCR5 expression on T cells in the T cell zone which may be disconnected from autoimmunity progression.
3. In vitro studies to assess Tfh interaction with B cells are not conclusive since activating naïve cells in the presence of basophils does not recapitulate Tfh developmental biology. In vivo systems and/or sorted Tfh cells must be used. Moreover, it is unclear in which context the basophils are being sorted. It is also unclear if the author's claim is that basophils control Tfh differentiation, expansion, maintenance or effector functions.
4. The authors suggest that IL4 produced by basophils leads to increases in PD-L1 on basophils. However, it is unclear how PD-L1 on basophils mediates Tfh cell expansion and/or effector function. Is it through PD-1? Is there a direct interaction in vivo? What are the consequences of this interaction?
5. The PD-L1 deletion phenotype is the most interesting portion of the manuscript but it is not explored in depth. The PD-L1 deletion on basophils was only performed in the context of pristane (and only 8 weeks after injection) and should also be studied in other more classical SLE systems (Sle1, Yaa, etc.) to make sure this phenotype is a general SLE phenotype and not an artifact of pristane. Comparison to other types of PD-L1 expressing cells (e.g. B cells) should also be performed to understand if any PD-L1 expressing cell can perform similar functions. Better quantification of PD-L1 deletion is also needed for both basophils, B cells and other APCs in Fig. S6 with repeats and statistics.
6. DT can affect B cell biology on its own which is not controlled for in these studies. Authors show one representative example that DT does not alter basophils but this should be reported with repeats and statistics.
7. Authors claim increased Tfh percentages in SLE which is believable (and shown previously) in terms of percentage of total CD4s, but most SLE patients are lymphopenic when they have a significant SLEDAI score. The authors should show total numbers of Tfh cells in SLE patients as well as in their basophil deletion/PD-L1cKO murine studies. Total numbers should also be given for basophils.
8. The authors rule out Tfr cells by stating the Tfr frequency of Tfh is the same with basophil deletion. However, as the total percentage of Tfh changes greatly there are likely alterations in the frequency of Tfr

of total CD4.

9. A possible explanation for the alteration in Tfh cells could be indirect through changes in GC B cells which are largely required for Tfh differentiation. Therefore, GC B cells should be assessed.

10. Plasmablast frequency does not necessarily correlate with disease in these autoimmunity models. The authors should assess these plasmablasts in more detail including antibody isotype by flow cytometry.

11. Comparison of OVA challenge to the phenotype in SLE is not a fair one. Methods suggests OVA challenge utilizes DT only during the last few days before harvest while lupus-like models have deletion for at least 10 days. The authors find basophils in spleen after ova challenge (albeit at a lower percentage compared to autoimmunity) and these cells are nicely deleted in their mice but there is no phenotype on the Tfh compartment. Is there a reason why basophils only have roles during autoimmunity?

Reviewer #3 (expert in systemic lupus erythematosus):

This is an interesting paper combining animal and human data which reports a novel function of basophils in promoting autoantibody production in systemic lupus erythematosus (SLE).

1. In the discussion authors need to make a more persuasive argument for the impact of their findings in SLE pathogenesis and the relative contribution of this mechanism to the disease.
2. Why the pristane model of SLE was used and not a spontaneous model? Please articulate more clearly

Point by point answers to reviewers' comments - NCOMMS-23-40765

We deeply thank the reviewers for their time and their challenging but very constructive comments. We hope that the answers and additional experiments that we provided addressed the reviewers' concerns.

REVIEWER 1: expert in basophils

INTRODUCTION

- "...basopenia that reflects their accumulation into SLO...". The finding of basopenia has been recently confirmed in pediatric SLE as well (see: *Diagnostics (Basel)*. 2022 Jul 12;12(7):1701. doi: 10.3390/diagnostics12071701; and, *Autoimmun Rev*. 2021 Apr;20(4):102790. doi: 10.1016/j.autrev.2021.102790). In detail, the first study above found some association with disease activity (SLEDAI), but not lupus nephritis. One point of the discussion was related to the sampling time point (in light of the ongoing therapy and previous disease course), which could be relevant also for the study participants enrolled in the present study.

In our previous and present studies, we analyzed three different adult SLE patient cohorts. We did find a clear association between basopenia and SLEDAI in the three cohorts (40 (*Nature Medicine* 2010); 222 (*Nature Communications* 2018) and 204 (present study)). As our main patient recruitment comes from our Nephrology department, the recruitment bias did not allow us to establish a specific association with lupus nephritis. We added the Spearman correlation and linear regression to the new supplementary Fig 1i showing in the presented cohort the (mild but significant) correlation between basopenia and disease activity.

Spearman r	
r	-0.3320
95% confidence interval	-0.4530 to -0.1989
P value	
P (two-tailed)	<0.0001
P value summary	****
Exact or approximate P value?	Approximate
Significant? (alpha = 0.05)	Yes
Number of XY Pairs	
	201

Equation	$Y = -4.199 * X + 22.05$
----------	--------------------------

We added as well the requested reference (*Diagnostics (Basel)*. 2022 Jul 12;12(7):1701).

We previously demonstrated that patients' treatment was not impacting the extent of the observed basopenia (*Nature Communications* 2018 Suppl Fig 1 see below). No patient under immunotherapy (rituximab or belimumab) were included.

(Pellefigues, Dema et al., *Nat Commun* 2018)

Concerning the sampling time point, inactive patients (SLEDAI 0/1) are mainly outpatients coming to the hospital for a usual care medical visit. The other patients are followed in the nephrology or in the Internal medicine department of the Bichat hospital and are seen by the physicians for the diagnosis of a flare (renal or not). The blood is drawn for analysis before any additional treatment is administered. For the nephrology department, after informed consent, blood is drawn and cells analyzed from all patients present at the hospital for a diagnostic kidney biopsy. The diagnostic is blinded to the operator and unblinded once the analysis are done and the cohort “frozen”. For the samples from the internal medicine department, the diagnostic was known.

- “In this context, targeting the addressing of basophils to SLO has demonstrated promising therapeutic potential.” The authors could clarify the therapeutic tools which demonstrated this potential.

Prostaglandin D2 receptors (PTGDRs) dual antagonism [laropiprant + CAY10471] (Nature Communications 2018) and AMG853 (Frontiers in Immunology 2022). These precisions are now indicated in the concerned sentence of the introduction.

In this regard, the authors could also mention the experience with omalizumab, which also alters basophil homeostasis by targeting IgE receptors (see: Arthritis Rheumatol. 2019 Jul;71(7):1135-1140. doi: 10.1002/art.40828)

This reference was added as well (it was already referenced/mentioned as we contributed to this trial).

- Other than these minor points, the introduction is clear and comprehensive.

METHODS

- “Blood samples were collected from adult patients enrolled in a prospective long-term study of systemic lupus erythematosus (SLE) and chronic renal diseases.” I would ask the authors to specify the study period.

The study period ranged from 04/2015 to 02/2020. The COVID19 pandemic/lockdown interrupted the recruitment. This information was added to the methods section.

- A specific ethical statement section could be created, including all the IRB specifications that the authors included in the mice section. Actually, specifications about approval for human and mouse studies should be clarified. As regards the IRB approval, the approval date should be also stated. In this section, clarifications about the study participants’ informed consent should be also included.

Reference ID-RCB 2014-A00809-38 (2014). The paragraph “Patient recruitment and ethics” was amended as requested.

- I would suggest to clearly separate the methods related to mice experiments and human analyses.

These paragraphs have been re-ordered.

- statistical methods: “When distribution were Gaussian or when N was too small,...”. Please, can you clarify when N is considered too small? Also, can you correct the grammar inconsistency?

We clarified and corrected the corresponding paragraph.

- statistical methods: “coupled with ad hoc multiple comparisons post-tests, as indicated in the figure legends depending on the distribution of the values in each group,...” Please, can you better clarify why different post hoc tests were used and, then, the criteria for this differential application?

We clarified the statistics section. We also re-did all statistical analysis to take into account all variables in the used tests (ie we performed two way ANOVA in the concerned graphical representations).

- For FACS analysis, especially in humans, can you clarify if these were performed on fresh samples or whether the biological samples were stored and analyzed in batch? In the latter case, can you provide the details?

Concerning the blood handling, “All samples were collected in heparinized tubes (BD vacutainer) and processed within 4 hours”. All data are from whole blood as visible in the gating strategy for basophils in Supp. Fig 1.

- As regards patients, can you clarify if specific inclusion and exclusion criteria were adopted to enroll them as study participants?

For the nephrology department, all patients coming for a diagnostic kidney biopsy and giving their consent were enrolled. Blood analysis was performed before the diagnostic was known. Physicians provided the diagnostics and clinical data afterwards. For the internal medicine department and outpatients from both departments, SLE patients were informed and proposed to participate. Exclusion criteria were limited to pregnancy and seropositivity for HIV, HBV or HCV to avoid any interference in the observed immunological phenotypes. This point was added to the corresponding section.

- "...patients' lupus activity who were classified as inactive (0-1), mild (2-4), and active (> 4)." I may suggest classifying patients as mildly active and moderately-severely active or anyway to revise this terminology. Then, it should be corrected in the figures and in the text accordingly.

A mention was added in the methods section "were classified as inactive (0-1), 'mild' for mildly active (2-4), and 'active' for moderately to severely active (> 4)." Adding the full proposed terminology to the figure themselves, although perfectly legitimate, may add some graphical confusion to the figures.

RESULTS

- "We and others reported that peripheral basopenia and activation of blood basophils correlate with disease activity in SLE patients". Notably, this association was found in the only study assessing basophils in pediatric SLE patients, which should be mentioned as well.

We did show this association in Nature Medicine 2010 (Fig 6d) and Nature Communications 2018 (Suppl. Fig 1f). However, as underlined by the reviewer, no correlation analysis was shown, but the extent of basopenia was significantly increased in groups with active disease. We added the indicated reference and the other study mentioning this parameter (Clin Rheumatol (2015) 34:891–896) in the corresponding paragraph and now show the correlation in the presented patient cohort between SLEDAI and absolute basophil numbers (New supp. Fig.S1i). Please see as well response to point #1.

- Following the last comment about the methods section, it would be important to clarify some clinical and demographic aspects of the patients included in this study, as well as the timing of the blood sampling, especially based on the clinical course and the ongoing/previous therapies. This information (which is included in the supplementary table 1) could be summarized in the first part of the results.

Please refer to the answer to the first point.

- In supplementary table 2, there is a column describing healthy controls. This controls should be defined in the methods section as well. I mean, who are they? How were they recruited?

These precisions have been added to the corresponding paragraph of the methods section.

- supplementary figure 1: panel F shows a covariation analysis and, in the caption, is reported as performed by Spearman coefficient. However, in the statistical methods, covariation analysis is not described at all. Please, complete the methods accordingly, also by defining the levels of correlation and statistical significance.

The statistics paragraph was modified accordingly.

- In supplementary figure 1, in several panels among b-e, the statistical significance between mild and inactive does not seem to be represented: can you clarify?

All the comparisons were added to the figure1 and S1.

- A similar clarification like the previous comment could be needed for figure 1. Following some previous comments about the classification of disease activity, one may guess that differences between all active SLE patients (mild + active) and inactive SLE patients could be not significant. Is this correct?

The requested comparisons were added to the Fig. 1 and Sup Fig S1 in all panels. When doing the comparisons between (mild+active) vs inactive, as anticipated by the reviewer, no significant differences are observed for TFH among CD4+ nor for TFH2 among CD4+. This point is part of the justification to separate inact./mild/active patients, the mildly active patients being either at the end of a previous flare, at the beginning of a new one or chronically mildly affected. Following reviewer #2 comments, we added the absolute numbers of TFH in our cohort in supplementary Fig. S1. Here, despite the CD4+ T cells lymphopenia in SLE patients, the absolute number of cTFH2 in active patients was significantly higher than mildly active patients. Moreover, as previously shown by others (Le Coz et al. PLoS One 2013), we added in Fig 1f the scattered plot showing a correlation between cTFH2 cell numbers and SLEDAI. Concerning basophils, (mild+active) vs inactive comparison is significant for CD203c and CXCR4 ($p < 0.001$), and for PD-L1 and CD84 ($p < 0.05$) on basophils.

- Is it possible to clarify in the text and/or figures how many patients were included in each group?

This mention was added in the legends of all figures.

- Similarly, as regards all murine experiment, I would suggest the authors to clarify the number of mice for each group, even if this could be probably understood from the figures itself. However, it would be more immediate if the authors disclose how many mice they used for each experiment/group.

These numbers were added in the figure legends.

- Beyond these minor aspects, I think that the results are explained well and with a clear logical flow.

DISCUSSION

- Are there any study limitations in the experimental model or, in general, in the study protocol, which may leave some doubts about the conclusions and main findings? If so, please, discuss and explain them.

A "limitations of the study" paragraph was added before the end of the discussion.

- In this regard, some specific discussion about the differences (and, then, limitation in the translational potential) between human SLE and murine model could be also included.

In the "limitations of the study" section, we indicated that results should be validated in SLO samples from patients, that represent a real technical and ethical challenge. However, in vitro results are even more potent in Human samples than in mouse samples on the ability of basophils to induce TFH cell differentiation suggesting an even more potent functional relationship. Further clinical development of approaches depleting basophils, IgE, IL-3 or antagonizing PTGDRs to prevent basophil accumulation in SLO will allow to validate functionally the basophil/TFH cell relationship in the human SLE context.

- At the beginning of the discussion, I would suggest the authors to schematically list the main and novel findings emerging from this study.

The short summary of the main findings was added to the beginning of the discussion.

- A very recent paper investigating the role of basophils and specific role of autoreactive IgE may further support and enrich the discussion (see: Rheumatology (Oxford). 2023 Feb 22;kead082. doi: 10.1093/rheumatology/kead082).

The paper by Fujimoto et al was already cited in the introduction. It reproduces some of our previous results (autoreactive IgE titers correlation with SLEDAI [Dema et al., PLoS One 2014] and basophil CD62L and CXCR4 overexpression leading to basophil accumulation in SLE patients secondary lymphoid organs [Charles et al., Nat Med 2010; Pellefigues, Dema et al., Nat Commun 2018]. We added a new reference to it in the discussion.

We deeply thank reviewer #1 for her/his time and constructive comments. We hope that the answers and additional analyses that we provided addressed the reviewer's concerns.

REVIEWER 2 – expert in TFH cells

In Tchen et al the authors are assessing the role of Basophils in controlling pathogenic Tfh responses during lupus-like disease. The authors find that Basophils in SLE patients express higher levels of PD-L1 compared to control patients. Utilizing pristane or Lyn^{-/-} models of autoimmunity, the authors find that basophils promote Tfh and IL-21 responses. In addition, the authors find that basophil PD-L1 has roles during basophil-mediated Tfh promotion. Overall, understanding the signals that promote pathogenic Tfh responses in SLE is an important topic.

However, this study lacks sufficient conceptual advance and mechanistic insights to be impactful. In particular, the authors have already demonstrated that basophils control SLE-like disease in vivo and the phenotype on Tfh cells is predictable since these cells promote autoantibodies.

Moreover, attempts at uncovering mechanisms are performed utilizing in vitro assays that do not recapitulate Tfh biology. More specific in vivo experiments need to be utilized to elucidate how PD-L1 on basophils may have roles in controlling pathogenic Tfh cells and SLE-like disease.

We thank the reviewer for her/his time and fruitful comments. We hope that the responses presented below will answer all the reviewer's concerns.

Major points

1. The authors have already shown in multiple publications that basophils promote autoreactive antibody and plasmablasts in pristane/Lyn^{-/-} autoimmunity models. The alteration in Tfh cells may be an indirect consequence of loss of pathogenic B cells and not necessarily due to an interaction with basophils. No mechanistic studies are performed in vivo to understand the nature of the basophil/Tfh interaction nor at which stages Tfh cells are altered. More sophisticated in vivo studies along with transcriptomics techniques are required to explore this in more detail.

We thank the reviewer for rising this important point. Additional experiments were done and further explanations are now provided in order to address this point on the effects of basophils on TFH functions. Fully differentiated GC TFH cells do not proliferate and localize exclusively in the GC area (and are CD90.2⁻). TFH cells (including GC TFH cells and GC TFH-like cells (CD90.2⁺)) promote B cell maturation inside (CD90.2⁻) and outside (CD90.2⁺) GC by providing IL-21 and IL-4 to their environment and interacting with activated B cells especially at the T cell:B cell border (CD90.2⁺) in spleen (Crotty Immunity 2019 PMID: 31117010 and Yeh et al., Immunity 2022 PMID: 35081372). GC TFH cells are needed for GC B cell survival and GC formation is, at least partially, dependent in activated B cell interactions with CD90.2⁺ GCTfh-like cells (Yeh et al., Immunity 2022). Depleting basophils in our lupus-like mouse models showed a decrease in switched B cells (Fig. 3), and a dramatic decrease in GC B cell proportions were observed in pristane-treated animals with IL-4 deficient basophils, PD-L1 deficient basophils or with constitutive basophil deficiency (Fig. S10). These data suggest that basophils control, in the lupus-like context, the ability of TFH cells to promote B cell maturation and GC structure maintenance. This is now further evidenced by the immunization approach when basophils are recruited in SLO (through PGD2 treatment, see point #11) and by the fact that the number of GC observed by microscopy is reduced in pristane-treated basophil deficient animals as compared to WT animals (Fig S5). We now show as well that basophils are mainly localized at the T cell:B cell border in spleen and have a dominant effect on CD90.2⁺ GC TFH-like cells, although they regulate as well the numbers of GC TFH cells.

We now provide as well some data from co-culture experiments where TFH cells have been purified from WT and Lyn^{-/-} mice basophil-depleted or not and cultured with naïve B cells from WT mice. Plasmablast differentiation was measured after three days of co-culture and we show that TFH cells from lupus-like and basophil sufficient mice are much more prone to induce plasmablast differentiation than TFH cells from lupus-like basophil-depleted (for 10 days) mice, demonstrating that basophil-induced TFH imprinting is really modulating TFH cell function in vivo beyond their effects on TFH cell numbers.

Finally, qPCR analyses of FACS-sorted TFH cells from lupus-like mice depleted or not from their basophils were realized (n=4-6 per group). The data presented below suggest that basophils promote the expression of Prdm1 in TFH cells in vivo but the other targets analysed were not modulated by basophil depletion in the Lyn^{-/-} model.

As Prdm1 expression by TFH cells and its role are not fully known yet (Miller et al., <https://doi.org/10.1101/2022.03.31.486642>), this point may be investigated deeper in future studies.

Of note, we wondered whether basophils were controlling CXCR5 expression levels on T cells that could have explained some of the observed phenotype, as suggested by the reviewer. As shown to the reviewer, no such effect was observed at the protein level (below) nor at the mRNA level (above).

Basophil depletion do not alter CXCR5 expression levels by TFH

(a,b) CXCR5 expression levels were quantified on spleen TFH cells by flow cytometry (ratio geometric mean fluorescence intensity (gMFI) of CXCR5 staining on gMFI of CXCR5 FMO) from (a) aged *Mcpt8*^{DTR} (blue) and basophil sufficient (DT-) or basophil-depleted (DT+) mice and from aged *Lyn*^{-/-} *Mcpt8*^{DTR} mice (red) DT-treated or not and from (b) PBS-injected (blue) and basophil sufficient (DT-) or basophil-depleted (DT+) mice and from pristane-injected *Mcpt8*^{DTR} mice (red) DT treated or not.

Altogether these new data strengthen and complete the data presented in the first version of the manuscript to push forward that basophils accumulate at the T:B border in the pristane-induced lupus-like disease, control TFH cell accumulation in SLO, their ability to produce IL-21 and IL-4 and their function in autoimmunity, have a dominant effect on CD90.2+ TFH cells outside GC and influence the number of GC in the lupus-like context.

2. The authors suggest the increase in Tfh cells upon basophil deletion is pathogenic because of small changes in IL-21 by flow cytometry, yet it is unclear if these Tfh cells are truly participating in autoimmunity progression.

The reviewer probably meant “the increase in TFH cells upon basophil influence” as basophil depletion leads to decreased TFH cell numbers in the lupus-like context.

We thank the reviewer for this comment as it underlines that our point was not explained clearly enough. The increase in TFH cell numbers is observed only in the lupus-like context (either *Lyn*^{-/-} background or pristane-induced lupus-like disease, thereafter called ‘PIL’, where basophils are accumulating in secondary lymphoid organs (see point #11). TFH cell accumulation has been demonstrated to be pathogenic in several lupus-like mouse models, including the PIL (Wang et al., JCI 2020, PMID: 32191636 and Faliti et al., J Exp Med 2019, PMID: 30655308). In our study, increased TFH numbers in lupus-like conditions are dependent on basophils (depletion experiments in *Lyn*^{-/-} and PIL and constitutive basophil deficiency in PIL (*Mcpt8*^{CT/+} *Rosa26-Stop*^{fl/fl} *DTA*) (Fig 2 and Fig. S3). Moreover, this increase in TFH numbers in the PIL model is dependent in PD-L1 and IL-4 expression by basophils (Fig. 5 and 6). On top of TFH cell number regulation, our study shows that basophils influence the TFH cell ability to produce IL-21 and IL-4 that are at the center of TFH cell functions (Crotty Immunity 2019). This is shown in the depletion experiments (Fig. 3) and now as well in the PIL in all our cKO mice (Fig. S9). To summarize, in the lupus-like context in vivo, basophils control TFH cell accumulation in SLO and their ability to produce IL-21 and IL-4 in a manner dependent on basophil-expressed PD-L1 and IL-4. These points are further supported by the point #1’s answer.

Of note, the basophil-dependent increase in TFH cells is linked to the autoimmune phenotype where basophils accumulate in SLO. This point will be further developed in the response to point #11. Basophil depletion leads to reduce TFH proportions back to WT (or PBS treated animals) levels but not to 0, meaning that “conventional” TFH cells do not seem to depend on basophils as shown in the immunization experiments (point #11). In other words, TFH cell expansion depends on basophils only in the lupus-like context (or when basophils are in SLO).

Basophils may be promoting CXCR5 expression on T cells in the T cell zone which may be disconnected from autoimmunity progression.

Basophil visualisation in SLO from PBS-treated and Pristane-treated animals, now shown in supplementary Fig. S5, finally define their localization at the T-B border as observed as well in the OVA immunization setting after PGD2 injection (cf. below).

CXCR5 expression by CD4+ T cells contributes to TFH retention and localization into GC-like structures (but is not mandatory) and contributes to the generation of GC B cells (Moriyama et al., J Exp Med 2014, PMID: 24913235). CXCR5^{hi}PD-1^{hi} CD4+ T cells in SLO can be classified as a mixture of GCTfh and GCTfh-like cells that have different TCR repertoires, different proliferation abilities (GCTfh-like cells proliferate, GCTfh cells do not) and different functions (Yeh et al., Immunity 2022, PMID: 35081372). The key marker to distinguish between those two TFH populations is CD90.2 (and S1PR2, but no antibody for flow cytometry is available). The new data presented in point#1 and in Fig. 3 underline the dominant effect of basophils on CD90.2+ TFH cells outside GC, at the T:B border. The localization of basophils in spleen during lupus-like disease by microscopy is now presented in Fig. S5.

3. In vitro studies to assess Tfh interaction with B cells are not conclusive since activating naïve cells in the presence of basophils does not recapitulate Tfh developmental biology. In vivo systems and/or sorted Tfh cells must be used.

This point has been addressed above (point #1, Fig. 3) with co-culture experiments between FACS-sorted TFH cells from *Lyn*^{-/-} mice depleted or not from basophils for 10 days and WT B cells. TFH from basophil-depleted *Lyn*^{-/-} mice were less efficient than TFH cells from basophil-sufficient *Lyn*^{-/-} mice to induce WT B cell differentiation into plasmablasts.

Moreover, it is unclear in which context the basophils are being sorted.

All basophils used in the co-culture experiments presented in Fig.4 were from CT-M8 mice (bred with the indicated floxed mice) and FACS-sorted based on the tomato expression. Purity check with antibodies >99%.

It is also unclear if the author's claim is that basophils control Tfh differentiation, expansion, maintenance or effector functions.

To summarize, basophils control TFH cell numbers in the lupus-like context in a PD-L1- and IL-4-dependent manner (expansion), TFH differentiation in vitro (human and mice Fig.4 and Fig. 8), maintenance in vivo (depletion approach in *Lyn*^{-/-}, PIL and OVA+PGD2 immunization cf. point #11 and Fig. 2, 3, 5 and 6), and effector functions (cytokine production by restimulation and constitutive production of cytokines in the lupus-like context; TFH ability to induce plasmablast differentiation and isotype switch in vivo (Fig. 3)).

4. The authors suggest that IL4 produced by basophils leads to increases in PD-L1 on basophils.

Ex vivo, unlike IL-3 and anti-IgE, we show that IL-4, does not induce PD-L1 overexpression by basophils but amplify the IL-3-induced PD-L1 overexpression (Fig. 4m). In the co-culture system, PD-L1 overexpression on basophils induced by CD4+ T cell-derived IL-3 is less efficient when basophils are IL-4 deficient (Fig. 4k). In vivo, if pristane treatment induces upregulated PD-L1 expression on basophils (Fig. 2 and 7a), it is much less efficient in pristane-treated *Mcpt8*^{CT/+} *Il4*^{fl/fl} mice (Fig. 7a). We concluded that basophil-derived IL-4 was amplifying IL-3 mediated PD-L1 upregulation on basophils in the lupus-like context.

However, it is unclear how PD-L1 on basophils mediates Tfh cell expansion and/or effector function.

Is it through PD-1?

In the co-culture system, when PD-1/PD-L1 interaction is blocked with the addition of a PD1-Fc construct, basophil-dependent TFH cell induction is inhibited (Fig. S7i). When basophils are PD-L1 deficient, they induce TFH cell differentiation much less efficiently than WT basophils (Fig. 4a and 4f). We could recapitulate basophil effects on TFH cell differentiation only by adding both IL-4 and PD-1 stimulating antibody (Fig. 4f,g,h,i). This set of data shows as well that PD-1 engagement on T cells is mandatory (together with IL-4) to induce Bach2 downregulation, Bcl6 maintenance, Prdm1/Maf/Batf/Gata3 upregulation and TFH cell differentiation.

Is there a direct interaction in vivo?

In Fig. S5, we now show that basophils localize mainly at the T-B border. We found few samples where basophils were inside GC, but it was clearly not representative of the majority of the observed localization in both OVA-PGD2 treated animals and pristane-treated animals. The direct interaction occurring in vivo was mainly observed at the T:B border and less frequently inside GC.

What is the consequences of this interaction?

Basophil-specific PD-L1 deficiency in vivo in the lupus-like context prevents IL-21 and IL-4 production amplifications by TFH cells (Fig. S9a,b) and TFH cell accumulation (Fig. 5). In vitro, the PD-L1/PD-1 dependent interaction promotes IL-21 production by TFH cells, and contributes to IL-6/4/13 production (Fig. 4b-e). Together with IL-4, it induces the downregulation of *Bach2* and the induction of key transcription factors (*Bcl6*, *Batf*, *Maf*, *Prdm1* and *Gata3*).

On the basophil side, this interaction induces both IL-4 and IL-6 production in vivo in the lupus-like context (Fig. 5c,d) and contribute to the IL-4 and IL-6 productions in the co-culture system (Fig 4 n-p).

5. The PD-L1 deletion phenotype is the most interesting portion of the manuscript but it is not explored in depth. The PD-L1 deletion on basophils was only performed in the context of pristane (and only 8 weeks after injection) and should also be studied in other more classical SLE systems (*Sle1*, *Yaa*, etc.) to make sure this phenotype is a general SLE phenotype and not an artifact of pristane.

Although mild, pristane after 8 weeks induces quantifiable pathogenic mechanisms (autoantibodies, IC glomerular deposits, plasmablast increase, TFH cell and basophil accumulations in SLO...). We recently demonstrated the nonredundant role of basophils in the pristane-induced lupus-like mouse model at 8 weeks after pristane injection (Tchen et al., *Frontiers Immunol* 2022). The results are the same 24 weeks post-injection as shown below, further validating the pristane-induced procedure at 8 weeks as relevant for the parameters analysed here. (added as Sup Fig. S3 in the new version).

Basophil-TFH functional relationship in lupus-like disease 24 weeks after pristane injection

(a,b,c) Proportions (%) of basophils among CD45⁺ splenocytes (a), of TFH among spleen CD4⁺ T cells (b), and of CD19⁺CD138⁺ cells among CD45⁺ splenocytes (c) from *Mcpt8*^{CT/+} (WT, blue) and basophil-deficient (*Mcpt8*^{CT/+} *R26*^{DTA/+}, red) mice treated with PBS or pristane (- or +, respectively) 24 weeks before analysis. (d,e,f) Proportions (%) of basophils among CD45⁺ lymph node (LN) cells (d), of TFH among LN CD4⁺ T cells (e), and of CD19⁺CD138⁺ cells among CD45⁺ LN cells (f) from the same mice as in (a,b,c). (g) Anti-RNP IgG plasma titers in mice as in (a,b,c) were quantified by ELISA. O.D. values at 450 nm were normalized to the mean of the PBS-injected mice of the same genotype. (a-g) Results are presented as individual values in bars representing the mean values ± s.e.m. (N = 3-5/group). Statistical analyses were done by two-way ANOVA followed by Tukey's multiple comparison tests between the indicated groups. NS: not significant, P>0.05; *: P<0.05; **: P<0.01; ***: P<0.001; ****: P<0.0001. A.U.: arbitrary units.

Although we do not have any doubt about the validity of the pristane approach, we agree with the reviewer that it may be of interest to reproduce the effects of the basophil-selective PD-L1 deficiency in another lupus-like mouse model. Since we don't have currently access to the suggested models (*Sle1*, nor TC (*Sle1.2.3*) nor NZB/NZW F1) bred with the *Mcpt8*-Cre-TdT (CT-M8) and with the *Pd1*^{fl/fl} mice, we added this point to the section discussing the limitations of the study that was suggested by reviewer #1.

Comparison to other types of PD-L1 expressing cells (e.g. B cells) should also be performed to understand if any PD-L1 expressing cell can perform similar functions. Better quantification of PD-L1 deletion is also needed for both basophils, B cells and other APCs in Fig. S6 with repeats and statistics.

We agree with the reviewer about the interest to decipher whether other PD-L1 expressing cells could perform functions similar to basophil ones on the TFH promotion in the lupus context. What we got from the literature is:

- 1- PD-L1 expression by CD11c-expressing cells inhibits TFH cell response in experimental autoimmune encephalomyelitis (PMID: 29531164).
- 2- PD-L1 at the surface of immature DC and monocytes from SLE patient is decreased and failed to be upregulated upon stimulation (PMID: 18650228) and proportions of PD-L1+ cells are decreased in PBMC from SLE patients (PMID: 35738802).
- 3- PD-L1 expression by CD11c+ Tbet+ B cells (age-related B cells and/or DN2 B cells) is increased in SLE patients and these cells are highly sensitive to IL-21 (PMID: 29717110).

As requested, we better quantified PD-L1 expression by other cell types in our pristane treated PDL1 cKO mice as now shown in Sup Fig. S8. In this new set of data we show that PD-L1 expression is increased on CD19+ B cells upon pristane treatment and not significantly increased on other antigen presenting cells (CD19- IA-IE^{hi} CD11c⁺). Importantly, the PD-L1 levels on those two cell types were not altered in our PDL1 cKO mice. Thus, based on this input, other PD-L1 expressing cells in SLO from PIL mice can not perform similar functions to basophils concerning the TFH cell accumulation in the SLE-like context.

6. DT can affect B cell biology on its own which is not controlled for in these studies. Authors show one representative example that DT does not alter basophils but this should be reported with repeats and statistics.

The reviewer probably meant “DT does not alter B cells” as DT induces basophil depletion in *Mcpt8^{DTR}* mice.

Pellefigues, Dema et al. Nat Communications 2018 PMID: 29463843 (supplementary Fig 5):

DT injection in WT or *Lyn*^{-/-} mice that do not have the DTR do not impact the B cell compartment at all as previously shown (Pellefigues, Dema et al., *Nat Commun* 2018, PMID: 29463843). Above: no impact on plasmablast nor autoantibody production. Concerning B cells in *Mcpt8DTR* mice, no direct effect of DT is observed in PBS-treated animals on CD19+ B cells (Fig. S2l,m) nor on myeloid cell compartments (Lamri et al., *J Allergy Clin Immunol* 2021, PMID: 33338538) unlike what was shown by El Hachem et al. (*EJ* 2020) who used *Mcpt8DTR* mice on a Balb/c genetic background (PMID: 29315532). A significant effect is observed only in pristane-treated animals (new supp fig. S2l and S2m) with absolute numbers of total spleen CD19+ B cells, suggesting an effect of basophil depletion, but not of DT itself.

7. Authors claim increased Tfh percentages in SLE which is believable (and shown previously) in terms of percentage of total CD4s, but most SLE patients are lymphopenic when they have a significant SLEDAI score. The authors should show total numbers of Tfh cells in SLE patients

As requested, we calculated the absolute numbers of CD4+ T cells, cTFH cells and cTFH2 cells in our patient samples. As anticipated by the reviewer, CD4+ lymphopenia was observed in our patients (Fig S1a). However, this did not lead to a cTFH decrease (Fig S1b) but to a normal number of cTFH cells. This point suggests that the change in the quality of CD4+ T cells (through an overrepresentation of cTFH) may have significant consequences in SLE patients. Importantly, absolute numbers of cTFH2 in the severely active patient group was significantly increased when compared to mildly active patients and correlated with disease activity (New Fig. 1f and supplementary Fig S1a-c). This point was reported in the text.

as well as in their basophil deletion/PD-L1cKO murine studies. Total numbers should also be given for basophils.

In human data, the absolute number of basophils was shown in Sup Fi. S1. For mouse studies, in depletion experiments, the absolute numbers of TFH cells and of basophils were added to the supplementary Fig. S2 (h-k) for both lupus-like models. We added as well these absolute numbers in the WT vs PD-L1cKO pristane models in the new supplementary Fig. S8 (e,f). This new information validates the effect of basophils on the accumulation of TFH cells in our models. We thank the reviewer for these very constructive suggestions.

8. The authors rule out Tfr cells by stating the Tfr frequency of Tfh is the same with basophil deletion. However, as the total percentage of Tfh changes greatly there is likely alterations in the frequency of Tfr of total CD4.

As requested, we did plot the proportions of TFR among CD4+ T cells (new supplementary Fig. S2g and see below) and found that proportions of TFR cells among CD4+ splenocytes were not significantly impacted by basophil depletion although a clear trend seemed to be present.

9. A possible explanation for the alteration in Tfh cells could be indirect through changes in GC B cells which are largely required for Tfh differentiation. Therefore, GC B cells should be assessed.

Recent and less recent reviews on TFH biology by S. Crotty underline the interdependency, in some settings, between GC B cells and GC TFH cells. If GC B cells are required for GC TFH cell differentiation, the opposite is true as well (Crotty ARI 2011, Crotty Immunity 2019). As underlined in these reviews, some models favour the fact that TFH cells are required for GC B cells differentiation and besides direct B-TFH contact, TFH cells also promote the generation of Abs with high affinity through soluble mediators like interleukin-4 and IL-21 outside GC (CD90.2+ GC TFH-like cells, Yeh et al., Immunity 2022).

As requested, we did analyse the GC B cells compartment in our models. Unfortunately, the CD95 marker was not included in the panels of most of our experiments, but as shown below in panel a, gating of CD19+ CD138- cells with GL7 and IA-IE (MHC II) gave almost the same population of cells (92% of identity). Based on this gating strategy, mice with basophil-selective IL-4 or PD-L1 deficiencies or constitutive basophil-deficiency did not have increased proportions of GC B cells (panel b).

Basophil-TFH relationship impacts GC B cell abundance in pristane-induced lupus-like disease

(a) Unconventional gating strategy was used to define germinal center (GC) B cells (CD45⁺CD19⁺CD138⁻GL7^{hi}IA-IE^{hi}) in spleen from the mice presented in (b). This gating strategy led to a 92% identity with the conventional gating of GC B cells (CD45⁺CD19⁺CD138⁻GL7^{hi}CD95⁺) as shown in the right panel. (b) Proportions (%) of GC B cells as defined in (a) among CD19⁺CD138⁻ splenocytes from mice *Mcpt8^{CT/+}* (WT), *Mcpt8^{CT/+}* *Ii4^{fl/fl}* (*Ii4^{fl/fl}*) (red), *Mcpt8^{CT/+}* *Pd1^{fl/fl}* (*Pd1^{fl/fl}*) (green) and basophil-deficient (*Mcpt8^{CT/+}* *R26^{DTA/+}*) (*R26^{DTA/+}*, grey) treated with PBS or pristane (- or +, respectively) for 8 weeks. (b) Results are presented as individual values in bars representing the mean values ($N = 3-14/\text{group}$). Statistical analyses were done by one-way ANOVA followed by Holm-Šidák's multiple comparisons tests between the indicated groups. NS: not significant, $P > 0.05$; ****: $P < 0.0001$.

The results are in line with the observed levels of anti-RNP IgG presented in the different figures of the manuscript (Fig. 5 and 6) and further indicate that the anti-RNP IgM detected in *Mcpt8^{CT/+}* *Ii4^{fl/fl}* pristane-treated mice are most likely produced by the extrafollicular pathway as discussed in the manuscript. This is further

evidenced by the effect of basophil deficiency and of basophil accumulation in SLO during OVA immunization (+PGD2) on the number of GC observed in immunized mice (see below, point #11).

10. Plasmablast frequency does not necessarily correlate with disease in these autoimmunity models. The authors should assess these plasmablasts in more detail including antibody isotype by flow cytometry.

In our previous studies, as in the present one, the autoreactive IgG antibody titers, the plasmablast frequencies and the glomerular IgG (and C3) deposits were and are always correlated in both lupus-like mouse models *Lyn*^{-/-} and pristane (Fig. 5 and Fig. 6). The only exception in our hands is the case of the *Mcpt8*^{CT/+} *II4*^{fl/fl} where we needed to consider the autoreactive IgM (Fig. 6). In the new experiments suggested by the reviewer (that complete the Fig. 3) we added an intracellular staining for IgM, IgG and IgA in plasmablasts:

Basophil depletion and plasmablast isotypes in *Lyn*^{-/-} lupus-like context

Mcpt8^{DTR} and *Lyn*^{-/-} *Mcpt8*^{DTR} mice were depleted from their basophils (+DT) or not (-DT) for ten days and splenocytes were analyzed for their content in plasmablasts of IgM, IgG and IgA isotypes. **(a)** *Left*. Gating strategy of plasmablasts (defined as CD19⁺CD138⁺ cells) gated as shown on living singlets CD45⁺CD4⁻ cells. *Right*. Intracellular staining for IgM, IgG and IgA are shown on CD19⁺CD138⁺ cells as on the left. Above each gate are shown the proportions (%) among CD45⁺ splenocytes of the indicated subset. **(b)** Proportions (%) of CD19⁺CD138⁺ cells (as defined in **(a)**) among CD45⁺ splenocytes from the indicated mice. **(c-e)** Proportions (%) of CD19⁺CD138⁺IgM⁺ **(c)**, CD19⁺CD138⁺IgG⁺ **(d)**, and CD19⁺CD138⁺IgA⁺ **(e)** cells among CD45⁺ splenocytes from the indicated mice. **(b-e)** Results are presented as individual values in bars representing the mean values ± s.e.m. (N = 4-6/group). Statistical analyses were done by two-way ANOVA followed by Tukey's multiple comparison tests between the indicated groups. NS: not significant, P > 0.05; #: P < 0.1; *: P < 0.05; **: P < 0.01.

In the manuscript, the levels of anti-RNP IgG and IgM in the various genotypes of the pristane-treated animals (Fig. 5 and 6) together with the IgG and C3 deposits are in line with what is suggested above. Of note, in the article by Roco et al., Immunity 2019 PMID: 31375460, the authors show that class switch recombination (CSR) occurs mainly outside GC (at the T:B border) before switched B cells migrate to the GC in an immunization setting. Basophils mainly localize at the T:B border in lupus-like conditions (Fig. S5 and response to next point) and have a dominant effect on CD90.2⁺ GC TFH-like cells (Fig. 3k-n, response to point #1 and Yeh et al. Immunity 2022). Thus, it may make sense that switched B cells which proportions depend on basophils in the lupus-like context (now presented in Fig. 3g) reflect as well the effect of basophil depletion on plasmablast isotypes shown above.

11. Comparison of OVA challenge to the phenotype in SLE is not a fair one. Methods suggests OVA challenge utilizes DT only during the last few days before harvest while lupus-like models have deletion for at least 10 days.

To clarify the procedure, we added the protocol of immunization/injections in the Fig. S4. The use of DT was limited to the last 2 days to make sure that the induced TFH were only due to OVA immunization and not to DT immunization. *Lyn*^{-/-} mice show increased TFH proportions in SLO independently of OVA immunization. In the draining lymph node, however, we can observe an additional TFH induction after OVA immunization (Fig. S4c).

The basophil depletion by DT in the *Lyn*^{-/-} mice for the last two days dampens TFH cells with or without OVA immunization of *Lyn*^{-/-} mice (Fig. S4) as it does in the lupus-like context after 10 days of basophil depletion (Fig. 2). However, this 2 days-long basophil depletion does not impact the TFH cells in the WT mice immunized with OVA. As you will see below, if we force basophil localization to SLO through PGD2 treatment of WT mice, then TFH cell induction is stronger and this supplementary TFH cell induction only is dependent on basophils (Fig. S4).

The authors find basophils in spleen after ova challenge (albeit at a lower percentage compared to autoimmunity) and these cells are nicely deleted in their mice but there is no phenotype on the Tfh compartment.

Basophils are found naturally in spleen of WT mice without any immunization at a proportion ranging from 0.05 to 0.2% of CD45⁺ splenocytes. In lymph nodes (LN), this proportion falls to less than 0.005% (0.05%) in an untouched WT C57BL6 mouse. OVA immunization alone does not induce basophil accumulation in neither spleen or LN (Fig S4 and Otsuka et al. Nat Commun 2013 PMID: 23612279) explaining why basophil depletion does not impact TFH cell response in this setting, unlike what is observed in the lupus-like context where basophils accumulate in both spleen and LN.

Is there a reason why basophils only have roles during autoimmunity?

In lupus-like condition, basophils are recruited and accumulate in SLO (spleen and LN). This is not happening during the OVA whole protein immunization (as shown), explaining why TFH proportions and OVA-specific IgG are not impacted by basophil depletion. Concerning the immunization in the lupus-like context, basophils are already accumulated in SLO where they will affect TFH biology in response to OVA. Thus, this functional relationship seemed to be linked to the accumulation of basophils in SLO observed in the lupus-like context.

We previously showed that intraperitoneal PGD2 injection in mice was leading to a CXCR4-dependent recruitment of basophils into SLO (Nat Commun 2018, PMID: 29463843). We then induced basophil accumulation in SLO in WT *Mcpt8*^{DTR}, *Mcpt8*^{CT/+} and *Mcpt8*^{CT/+} *Rosa26*^{DTA/+} (basophil deficient) mice by injecting PGD2 to the mice every three days during the whole OVA immunization procedure (Fig. S4a). As expected, this led to the accumulation of basophils in spleen and draining (mesenteric) lymph node. The accumulation of basophils occurred mainly at the T-B border in the spleen, and resulted in increased germinal center (GC) formation as compared to non-PGD2 treated OVA-immunized mice (Fig. S4 and S5a-d). Importantly, promotion of GC formation was not observed in PGD2-treated OVA-immunized basophil-deficient animals further supporting the basophil contribution to this phenomenon (Fig. S5a-d). Mimicking the lupus-like context through basophil accumulation in SLO, PGD2 treatment made the TFH expansion and the resulting anti-OVA IgG production dependent on basophils as shown after DT-induced basophil depletion (Fig. S4). Of note, spleen basophils of pristane-treated animals were as well located mainly at the T-B border and were needed to observe increased GC-like structure accumulation and lupus-like disease onset (Fig. S5e-g). These new data further validate the functional relationship between TFH cells and basophils in SLO where they are accumulated.

In the discussion, we mentioned previously the observed phenotype in the literature of basophils during SARS-Cov2 infection which is very similar to what we see in SLE patients (basopenia suggesting recruitment to SLO, CD62L & CXCR4 & PD-L1 overexpression on basophils, increased IL-3 and CXCL12 titers...) and this phenotype is associated with a good (but transient) humoral response to the virus and a positive outcome for the patients. These features may indicate that in the context of a viral infection, basophils may be recruited as well to the SLO and, there, influence the antibody response through their effects on TFH. This point is further supported by the fact that severe COVID-19 patients are more prone to develop an autoimmune condition within three months post-infection (Mageau et al. J Intern Med. 2024, PMID: 38064539). This hypothesis will be developed in future studies.

We deeply thank the reviewer #2 for her/his challenging but very constructive comments. We hope that the answers and additional experiments that we provided addressed the reviewer's concerns.

Reviewer #3 (expert in systemic lupus erythematosus):

This is an interesting paper combining animal and human data which reports a novel function of basophils in promoting autoantibody production in systemic lupus erythematosus (SLE).

1. In the discussion authors need to make a more persuasive argument for the impact of their findings in SLE pathogenesis and the relative contribution of this mechanism to the disease.

We added the requested arguments to the discussion. To briefly summarize here: basophils are activated in SLE patients and lupus-like mouse models by autoreactive IgE-containing immune complexes and IL-3 (previous studies by us and others). PGD2 levels in SLE patient blood are elevated and induce CXCR4 externalization on basophils, making them sensitive to CXCL12 gradients arising from SLO where they will be recruited. There, they promote plasmablast accumulation and autoantibody production (PMID: 29463843). Here, we show how basophils actually do in SLO to promote autoantibody production. They control pathogenic TFH cell accumulation and TFH2 bias that occur during lupus pathogenesis through the expression of PD-L1 and IL-4. Thus, basophil depletion or prevention of their accumulation in SLO during lupus would break the basophil-dependent autoantibody production amplification and represent promising therapeutic strategies that should be developed.

2. Why the pristane model of SLE was used and not a spontaneous model?

Our study used two lupus-like mouse models, including one spontaneous model with the *Lyn*^{-/-} mice where we show as well the basophil-dependent TFH cell accumulation. *Lyn* deficiency induces a phenotype that is closed to the *FcyRIIb*^{-/-} mice phenotype. The pristane inducible model (pristane-induced lupus-like disease (PIL)) shares many features of the human disease, including type I IFN signature (Reeves et al., Trends Immunol 2009, PMID: 19699150). The PIL offers the advantage of being independent of any genetic bias other than the genetic background of the injected mice. We previously showed that basophils were contributing to the PIL (Dema et al., Sci Rep 2017, PMID: 28801578) and were having a nonredundant role in this model (Tchen et al. Frontiers Immunol 2022, PMID: 35844602). As our constitutive basophil-deficient mice (*Mcpt8*^{CT/+} *Rosa*^{DTA/+}) and mice deficient selectively in the basophil compartment for IL-4, for IL-6 or for PD-L1 (*Mcpt8*^{CT/+} *Il4*^{fl/fl}, *Mcpt8*^{CT/+} *Il6*^{fl/fl}, *Mcpt8*^{CT/+} *Pdl1*^{fl/fl}, respectively) were available, we investigated our working hypotheses with the PIL. As it is mentioned now in the discussion, basophil-selective PD-L1 deficiency in a more classical and severe lupus-like model (as NZBxNZW F1, MRL-*Fas*^{lpr} or B6.NZM Sle1/Sle2/Sle3) would validate our findings, but would require a minimum of 3 years to get *Mcpt8*^{CT/+} *Pdl1*^{fl/fl} mice on a pure NZB background and on a pure NZW background or on the MRL-*Fas*^{lpr} genetic background and more to get the triple congenic mice (TC Sle1/2/3 on the *Mcpt8*^{CT/+} *Pdl1*^{fl/fl} background).

Please articulate more clearly

We thank the reviewer #3 for her/his time and comments. We hope that the answers provided addressed the reviewer's concerns.

REVIEWERS' COMMENTS

Reviewer #1 (Remarks to the Author):

I thank the authors for their clarifications. I have no additional major points to address and/or clarify.

Reviewer #2 (Remarks to the Author):

I appreciate the amount of work the authors have performed for the revised manuscript. The authors have addressed many of my concerns with additional experimentation (or rationale that experiments would take too long), however a few concerns remain:

1. Although Garnett Kelsoe's work does show that CD90.2- Tfh cells are perhaps the real GC-resident Tfh cells, this has not been widely accepted/validated in all systems and many labs still utilize CXCR5hi expression as GC Tfh. Authors should temper their claims of this as it is possible that alteration of basophils/cytokines in autoimmunity may alter CD90 expression in Tfh independently of localization. Authors should also put the CXCR5 expression data (generated in previous comment #1 of point by point) into the main Figure 3. This is particularly important as the authors did not assess GC Tfh cells by microscopy.
2. Fig. 4h utilizes a "PD-1 stimulating antibody", yet details are missing from the methods. Which antibody is this and how do the authors know it is truly driving a signal through PD-1?
3. The concern that DT treatment was altering responses was not sufficiently addressed. I appreciate the authors pointing to experiments in S2I,m showing that DT does not alter Tfh cells in non-lupus settings, however the concern is that DT can alter responses in lupus settings. The authors should perform PBS vs. DT treatment in WT pristane induced mice in a similar timeline as in Fig. 3.

Reviewer #3 (Remarks to the Author):

No further comments

Point by point answers to referees' comments

NCOMMS-23-40765A

We deeply thank the reviewers for their time and their challenging but very constructive and positive comments. We hope that we addressed all reviewers' concerns.

REVIEWERS' COMMENTS

Reviewer #1 (Remarks to the Author):

I thank the authors for their clarifications. I have no additional major points to address and/or clarify.

Reviewer #2 (Remarks to the Author):

I appreciate the amount of work the authors have performed for the revised manuscript. The authors have addressed many of my concerns with additional experimentation (or rationale that experiments would take too long), however a few concerns remain:

1. Although Garnett Kelsoe's work does show that CD90.2- Tfh cells are perhaps the real GC-resident Tfh cells, this has not been widely accepted/validated in all systems and many labs still utilize CXCR5hi expression as GC Tfh. Authors should temper their claims of this as it is possible that alteration of basophils/cytokines in autoimmunity may alter CD90 expression in Tfh independently of localization. Authors should also put the CXCR5 expression data (generated in previous comment #1 of point by point) into the main Figure 3. This is particularly important as the authors did not assess GC Tfh cells by microscopy.

The CXCR5 expression data have been added to the main Figure 3 and the main text amended as follow: "Whereas its high expression may also reflect their localization into GC, CXCR5 levels on TFH cells were not modified by basophil depletion in both lupus-like mouse models (Fig. 3o,p)".

2. Fig. 4h utilizes a "PD-1 stimulating antibody", yet details are missing from the methods. Which antibody is this and how do the authors know it is truly driving a signal through PD-1?

We realized that the anti-PD1 antibody used was not listed in the table S3. We apologize for this omission. It has been added (Purified Anti-mouse CD279 (PD-1) (Clone: 29F.1A12); Biolegend; (Cat# 135202, RRID:AB_1877121)). The technical datasheet of the product does not indicate a "stimulating" feature of the antibody but a PD-1-ligands interactions blocking capacity. The "stimulating" term was used in the response to referees' comments and not in the main text and was obviously a mistake. However, as this antibody is coated on culture plates and recognizes PD-1, we can assume that it drives PD-1 crosslinking in the culture conditions. This point is further supported by its functional effects on TFH differentiation ex vivo presented in Fig. 4h/i, especially when combined with soluble IL-4.

3. The concern that DT treatment was altering responses was not sufficiently addressed. I appreciate the authors pointing to experiments in S2l,m showing that DT does not alter Tfh cells in non-lupus settings, however the concern is that DT can alter responses in lupus settings. The authors should perform PBS vs. DT treatment in WT pristane induced mice in a similar timeline as in Fig. 3.

We indicate here that the editorial board answered to this comment:

Editor: "Please note we do not require additional experiments for Comment 3 from Reviewer 2. We have made the decision that the data shown in the rebuttal and in supplementary Fig, 2l and m sufficiently demonstrate that DT does not alter the B cell compartment (including in lupus)."

Reviewer #3 (Remarks to the Author):

No further comments